# Structural basis for AcrVA4 inhibition of specific CRISPR-Cas12a

Gavin J Knott[1], Brady F Cress[1], Jun-Jie Liu[1], Brittney W Thornton[1], Rachel J Lew[2], Basem Al-Shayeb[3], Daniel J Rosenberg[4,5], Michal Hammel[4], Benjamin A Adler[6,7], Marco J Lobba[8], Michael Xu[1], Adam P Arkin[7,9], Christof Fellmann[2,10], Jennifer A Doudna[1,2,4,8,11,12,13]*

[1]Department of Molecular and Cell Biology, University of California, Berkeley, Berkeley, United States; [2]Gladstone Institutes, San Francisco, United States; [3]Department of Plant and Microbial Biology, University of California, Berkeley, Berkeley, United States; [4]Molecular Biophysics and Integrated Bioimaging Division, Lawrence Berkeley National Laboratory, Berkeley, United States; [5]Graduate Group in Biophysics, University of California, Berkeley, Berkeley, United States; [6]UC Berkeley-UCSF Graduate Program in Bioengineering, University of California, Berkeley, Berkeley, United States; [7]Department of Bioengineering, University of California, Berkeley, Berkeley, United States; [8]Department of Chemistry, University of California, Berkeley, Berkeley, United States; [9]Environmental Genomics and Systems Biology Division, Lawrence Berkeley National Laboratory, Berkeley, United States; [10]Department of Cellular and Molecular Pharmacology, University of California, San Francisco, San Francisco, United States; [11]Innovative Genomics Institute, University of California, Berkeley, Berkeley, United States; [12]Howard Hughes Medical Institute, University of California, Berkeley, Berkeley, United States; [13]California Institute for Quantitative Biosciences (QB3), University of California, Berkeley, Berkeley, United States

*For correspondence:
doudna@berkeley.edu

**Abstract** CRISPR-Cas systems provide bacteria and archaea with programmable immunity against mobile genetic elements. Evolutionary pressure by CRISPR-Cas has driven bacteriophage to evolve small protein inhibitors, anti-CRISPRs (Acrs), that block Cas enzyme function by wide-ranging mechanisms. We show here that the inhibitor AcrVA4 uses a previously undescribed strategy to recognize the *L. bacterium* Cas12a (LbCas12a) pre-crRNA processing nuclease, forming a Cas12a dimer, and allosterically inhibiting DNA binding. The *Ac. species* Cas12a (AsCas12a) enzyme, widely used for genome editing applications, contains an ancestral helical bundle that blocks AcrVA4 binding and allows it to escape anti-CRISPR recognition. Using biochemical, microbiological, and human cell editing experiments, we show that Cas12a orthologs can be rendered either sensitive or resistant to AcrVA4 through rational structural engineering informed by evolution. Together, these findings explain a new mode of CRISPR-Cas inhibition and illustrate how structural variability in Cas effectors can drive opportunistic co-evolution of inhibitors by bacteriophage.

DOI: https://doi.org/10.7554/eLife.49110.001

## Introduction

Biological warfare between microbes and bacteriophage drives the co-evolution of diverse host-phage interactions. Clustered regularly interspaced short palindromic repeats (CRISPR) and CRISPR-associated (Cas) proteins provide adaptive immunity in which Cas nucleases are deployed together

with CRISPR RNAs (crRNAs) to base-pair with foreign genetic material and trigger its destruction (*Knott and Doudna, 2018*). The potency of CRISPR-Cas immunity drives mobile genetic elements to evolve mechanisms that enable escape from CRISPR-Cas targeting. While genetic diversity offers some advantages when facing innate immune systems (*Labrie et al., 2010*), nucleotide variation alone is insufficient to evade the adaptability of CRISPR-Cas (*van Houte et al., 2016*). To avoid CRISPR-Cas targeting, many bacteriophage employ small anti-CRISPR (Acr) proteins to inactivate specific CRISPR-Cas systems (*Borges et al., 2017*; *Stanley and Maxwell, 2018*). Since the initial discovery of type I-F CRISPR-Cas inhibitors (*Bondy-Denomy et al., 2013*), a wide diversity of viral proteins have been found to inhibit CRISPR-Cas enzymes. Furthermore, genomic data analysis suggests that unique Acrs may exist for the majority of CRISPR-Cas subtypes (*Watters et al., 2018*).

Cas12a, like Cas9, is a biotechnologically relevant CRISPR-Cas enzyme for which novel Acrs have been recently identified (*Watters et al., 2018*). Cas12a (formerly Cpf1) is an RNA-guided Class II type V-A CRISPR nuclease which cleaves DNA following recognition of a 20-nucleotide DNA sequence containing a protospacer-adjacent motif (PAM) (*Zetsche et al., 2015*) (*Figure 1A*). Unlike Cas9, Cas12a directly catalyzes precursor crRNA (pre-crRNA) processing to generate a mature surveillance complex whose crRNA bears a hydroxyl on its 5' terminus (*Zetsche et al., 2015*; *Fonfara et al., 2016*; *Swarts et al., 2017*) (*Figure 1A*). Previously identified inhibitors of Cas12a, known as type V-A Acrs (AcrVA), either reversibly block substrate association through stoichiometric binding or irreversibly inactivate the enzyme complex through a chemical transformation (*Suresh et al., 2019*; *Knott et al., 2019*; *Dong et al., 2019*). While exhibiting stoichiometric inhibition of Cas12a, the inhibitor AcrVA4 appears to be unique in several ways. At 234 amino acids, AcrVA4 is almost twice as large as other anti-CRISPR proteins and it contains a predicted coiled-coil domain not observed in other Acrs. In addition, similar to specific Acrs that target Cas9 (*Harrington et al., 2017*) or Cas10 (*He et al., 2018*), its mode of action appears to involve dimerization of Cas12a (*Knott et al., 2019*).

We show here that AcrVA4 is a robust inhibitor of some Cas12a homologs but remains incapable of inhibiting other closely related homologs. Using single particle cryo-electron microscopy (cryo-EM), we find that AcrVA4 recognizes the conserved pre-crRNA processing nuclease of Cas12a and the specific chemistry of a mature crRNA bound within the enzyme. This specific association serves to position AcrVA4 proximal to highly conformationally dynamic domains that are locked by the inhibitor to cage the enzyme in a state incompatible with dsDNA binding. Structure-guided mutagenesis revealed that AcrVA4 dimerization is not required for inhibition of Cas12a in vitro and provided no advantage to bacteriophage targeted by Cas12a in vivo. Finally, we identified the structural basis for AcrVA4 ortholog specificity and engineered the AsCas12a enzyme to be susceptible to AcrVA4 in phage assays and human cell editing experiments. These results reveal a new mode of inhibition by anti-CRISPRs and demonstrate how structural variability in Cas effectors can drive opportunistic co-evolution of inhibitors by bacteriophage.

## Results

### The AcrVA4 C-terminal domain binds to LbCas12a-crRNA

Despite its distinction as one of the largest bacteriophage-derived protein inhibitors of CRISPR-Cas, AcrVA4 has a narrow spectrum of Cas12a inhibition specificity. Previous biochemical experiments demonstrated that AcrVA4 blocks dsDNA binding to *L. bacterium* (Lb) and *M. bovoculi* (Mb) Cas12a but not *Ac. species* (As) Cas12a (*Watters et al., 2018*; *Knott et al., 2019*), the last of which is commonly utilized as a tool for genome editing (*Figure 1B*). To investigate the mechanism of AcrVA4-mediated Cas12a inhibition and identify the basis for its ortholog specificity, we determined the structure of the LbCas12a-crRNA complex bound to AcrVA4 using single-particle cryo-EM (*Figure 1C–D*, *Table 1*, *Figure 1—figure supplement 1A*). From a subset of 156,979 particle images identified by three-dimensional classification, we reconstructed a three dimensional cryo-EM density map at 3.0 Å resolution (State I) (*Figure 1—figure supplement 1A*). This high-resolution map contained clear density into which the previously determined structure of the LbCas12a-crRNA complex was docked and refined (*Figure 1C–D*). With the exception of the PAM-interacting (PI) domain, which was disordered in our structure, we resolved most of the LbCas12a protein and crRNA to high-resolution (*Figure 1C–D*, *Figure 1—figure supplement 2A–B*). The map contained additional

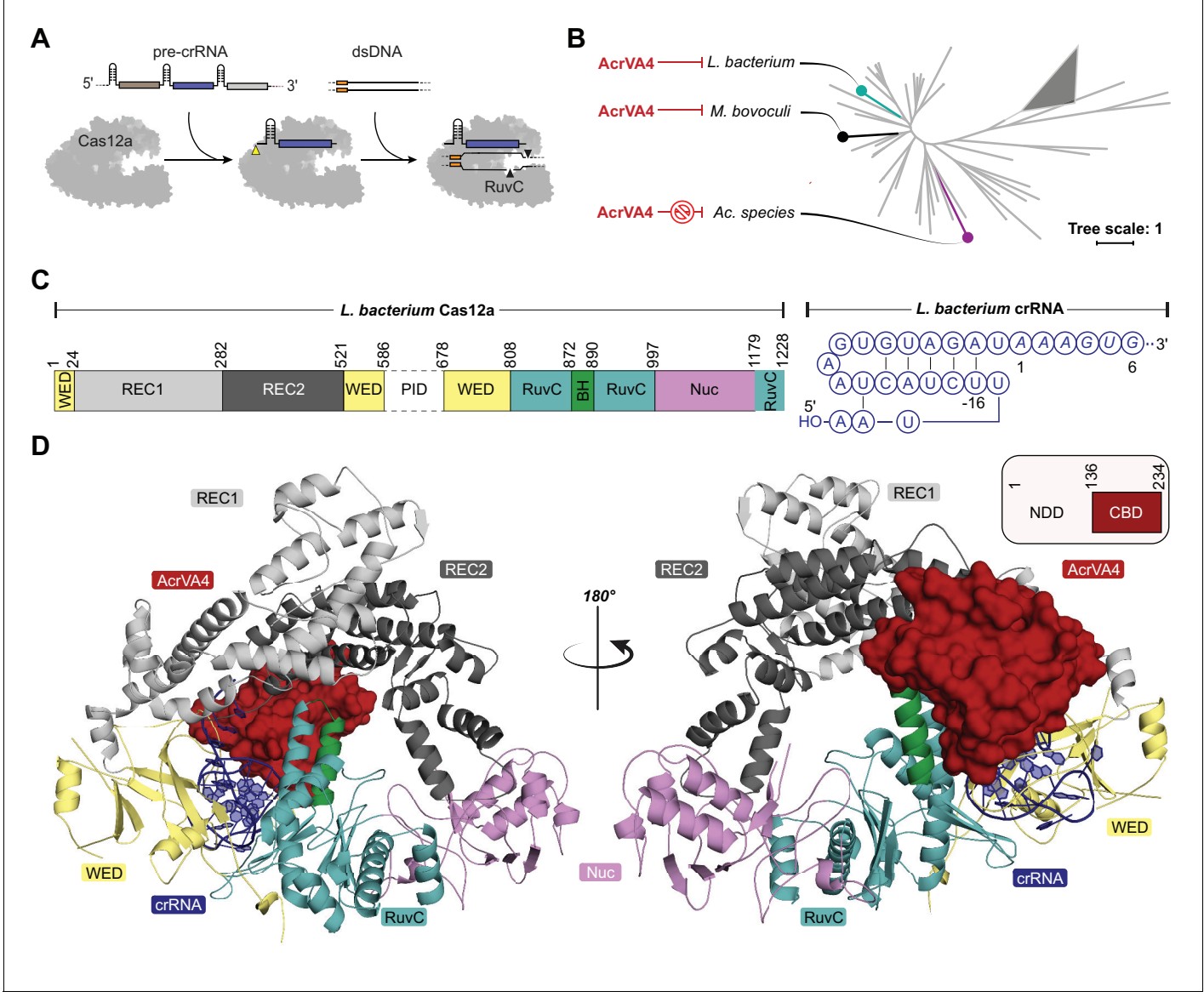

**Figure 1.** Overall structure of the LbCas12a-crRNA-AcrVA4 complex. (**A**) Schematic representation of Cas12a activity. (**B**) Unrooted maximum likelihood phylogenetic tree of Type V-A CRISPR-Cas12a. Species known to be susceptible or unsusceptible to phage-derived AcrVA4 are highlighted. The triangle denotes collapsed branches of Cas12b-e. (**C**) Schematic representation of LbCas12a and the mature crRNA modeled within the cryo-EM structures. (**D**) Two views of the LbCas12a-crRNA complex (cartoon) bound to AcrVA4 (surface) shown related by a 180° rotation. The color scheme for the crRNA, LbCas12a, and AcrVA4 in panels B, C, and D are used throughout the manuscript.

DOI: https://doi.org/10.7554/eLife.49110.002

The following figure supplements are available for figure 1:

**Figure supplement 1.** Cryo-EM data collection and 3D reconstruction.

DOI: https://doi.org/10.7554/eLife.49110.003

**Figure supplement 2.** Cryo-EM density for State I.

DOI: https://doi.org/10.7554/eLife.49110.004

**Figure supplement 3.** Structurally similar proteins to AcrVA4 CBD as determined by Dali search.

DOI: https://doi.org/10.7554/eLife.49110.005

density not attributable to LbCas12a-crRNA that allowed for unambiguous de novo tracing and sequence register assignment of the C-terminal domain of AcrVA4 (residues 136–233) (*Figure 1D*, *Figure 1—figure supplement 2C*).

**Table 1.** Cryo-EM data collection, reconstruction, and model statistics.

| | State I | State II |
|---|---|---|
| | PDB: 6P7M \| EMDB: 20266 | PDB: 6P7N \| EMDB: 20267 |
| **Data Collection** | | |
| Microscope | FEI Titan Krios | FEI Titan Krios |
| Voltage (kV) | 300 | 300 |
| Camera | Gatan K2 Summit | Gatan K2 Summit |
| Defocus range (μm) | 0.7 ~ 2.2 | 0.7 ~ 2.2 |
| Pixel Size (Å) | 0.9 | 0.9 |
| Magnification | 135000 | 135000 |
| Electron Dose (e/Å$^2$) | 47 | 47 |
| Total Particles | 324336 | 324336 |
| **Reconstruction** | | |
| Software | CryoSparc | CryoSparc |
| Symmetry Imposed | C1 | C1 |
| Final number of refined particles | 156979 | 79787 |
| Resolution of polished unmasked map (Å) | 4.2 | 8.7 |
| Resolution of polished masked map (Å) | 3 | 5 |
| Map Sharpening B-factor (Å$^2$) | −101.4 | −123.5 |
| **Refinement** | | |
| Model Resolution cutoff (Å) | 3 | 5 |
| FSC threshold | 0.143 | 0.143 |
| Map CC (box) | 0.74 | 0.78 |
| Map CC (mask) | 0.79 | 0.51 |
| **R..m.s Deviations** | | |
| Bond lengths (Å) | 0.006 | 0.002 |
| Bond angles (°) | 0.607 | 0.524 |
| Molprobity score | 1.77 | 1.84 |
| Clashscore | 7.87 | 6.63 |
| Rotamer Outliers (%) | 0 | 1.33 |
| **Ramachandran plot** | | |
| Favored (%) | 94.43 | 94.43 |
| Allowed(%) | 5.57 | 5.57 |
| Outliers (%) | 0 | 0 |

DOI: https://doi.org/10.7554/eLife.49110.006

Residues 136–233 of AcrVA4 adopt a compact $\alpha_1\beta_1$-$\beta_5\alpha_2$ fold, defined here as the C-terminal binding domain (CBD), that forms an extensive interface with LbCas12a (*Figure 1D*, *Figure 1—figure supplement 2C*). The 1479 Å$^2$ of buried surface area between AcrVA4 and LbCas12a accounts for 23.8% of the total solvent accessible surface area of the CBD. A Dali search (*Holm and Rosenström, 2010*) of the AcrVA4 CBD detected very limited structural similarity to the *Pyrobaculum* spherical virus (PDB code: 2X5C), the TrmB archaeal transcriptional regulator (PDB code: 3QPH), and several PAZ domains from Argonaute proteins (PDB code: 4Z4H) (*Figure 1—figure supplement 3*). While the CBD of only a single copy of AcrVA4 was well resolved in State I, AcrVA4 exists as an obligate dimer in solution that binds directly to either one or two copies of the LbCas12a-crRNA complex (*Knott et al., 2019*). Examining the EM density of our 3.0 Å reconstruction revealed a smearing of poorly defined density adjacent to the CBD (*Figure 1—figure supplement 1A*) most

likely attributable to a flexible protomer of AcrVA4 in the absence of another LbCas12a-crRNA protomer.

## AcrVA4 recognizes the Cas12a processing nuclease and crRNA 5'-OH

The CBD of AcrVA4 is nestled on the surface of the LbCas12a-crRNA complex wedge (WED) domain and abutted against the recognition 1 (REC1), recognition 2 (REC2) and RuvC domains (*Figure 2A*). The beta-stranded body of AcrVA4 sits directly on the solvent exposed surface of the LbCas12a processing nuclease, forming a number of charged or polar contacts. AcrVA4 inserts a glutamate (VA4:E178) to interact with the 5'-hydroxyl of the processed crRNA and a conserved histidine (Cas12a:H759) within the WED domain (*Figure 2B*). These interactions are further stabilized by T201 (*Figure 2B*) and an extended series of main chain interactions across the WED domain (*Figure 2— figure supplement 1*). To test the importance of residues contributed by AcrVA4 in this interaction, we substituted AcrVA4 residues E184 or T201 for alanine and assayed inhibition of LbCas12a-cata-lyzed dsDNA *cis*-cleavage. While the substitutions did not compromise the fold or oligomerization state of the inhibitor (*Figure 2—figure supplement 2A*), we observed that both individual AcrVA4 point mutations reduced the inhibition of LbCas12a activity (*Figure 2C*). The interface between LbCas12a and the CBD of AcrVA4 extends beyond the conserved crRNA processing nuclease to

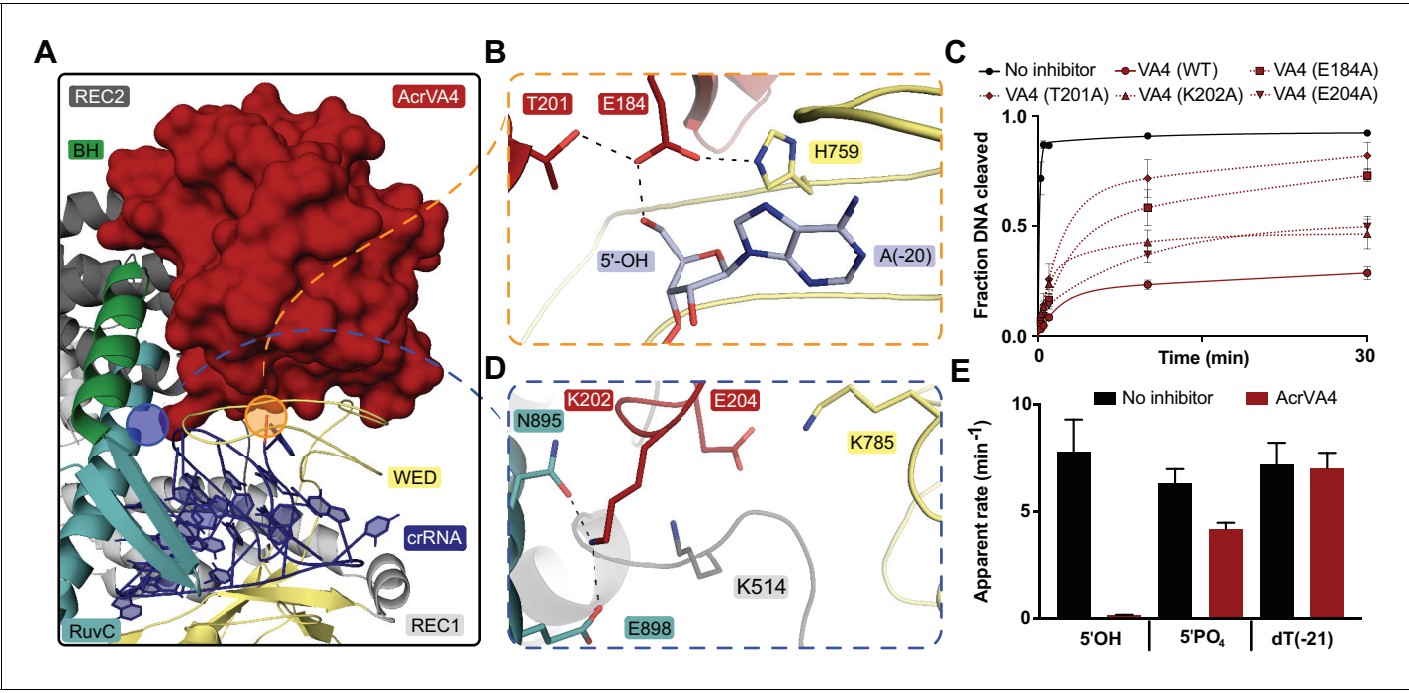

**Figure 2.** AcrVA4 recognizes the crRNA 5'-OH and Cas12a crRNA processing nuclease. (**A**) AcrVA4 (surface) is shown on the LbCas12a WED domain (cartoon, yellow) and crRNA (blue, cartoon) braced against the bridge-helix (green, cartoon), RuvC (teal, cartoon), REC1 (cartoon, gray), and REC2 domains (cartoon, dark gray). (**B**) Detailed atomistic view of AcrVA4 (red) recognition of the WED domain pre-crRNA processing nuclease (yellow) and crRNA 5'OH (blue). (**C**) LbCas12a dsDNA *cis*-cleavage over time measured under single-turnover conditions in the presence or absence of AcrVA4 containing alanine substitutions (mean ∓ s.d., $n = 3$ independent measurements). Two-phase exponential decay experimental fits are shown as solid or dashed lines. (**D**) Detailed atomistic view of the AcrVA4 (red) interface with the LbCas12a RuvC (teal), REC2 (dark gray), and WED domains (yellow). (**E**) Bar-graph illustrating the apparent rate of LbCas12a-mediated dsDNA *cis*-cleavage under single-turnover conditions guided by a crRNA bearing a 5'-OH, 5'-PO$_4$ or 5'-dT (−21) in the presence or absence of AcrVA4 (mean ∓ s.d., $n = 3$ independent measurements).

DOI: https://doi.org/10.7554/eLife.49110.007

The following figure supplements are available for figure 2:

**Figure supplement 1.** Extended interaction network between the AcrVA4 CBD and the LbCas12a-crRNA complex WED domain.

DOI: https://doi.org/10.7554/eLife.49110.008

**Figure supplement 2.** Size exclusion chromatography (SEC) traces for AcrVA4 mutants used in this study and the effect of 5'-crRNA chemistry on AcrVA4 binding to LbCas12a.

DOI: https://doi.org/10.7554/eLife.49110.009

include an elongated $\beta_3$-$\beta_4$ loop of AcrVA4 that sits deep in a solvent accessible pocket formed between the crRNA hairpin, REC2 and RuvC domains (*Figure 2A,D*). AcrVA4 contributes K202 from the $\beta_3$-$\beta_4$ loop into a negatively charged pocket formed between N895 and E898 of the RuvC domain (*Figure 2D*). This interaction is proximal to another charged interaction in which E204 of AcrVA4 is buried within a positively charged pocket formed by K514 and K785 between Cas12a's REC2 and WED interface (*Figure 2D*). To explore the importance of these interactions, we generated K202A and E204A mutants of AcrVA4 and found that they displayed a reduced ability to inhibit LbCas12a-catalyzed dsDNA *cis*-cleavage (*Figure 2C*, *Figure 2—figure supplement 2A*), consistent with their contributions to the binding interface between Cas12a and AcrVA4 (*Figure 2D*). Taken together, these results suggested that the bacteriophage-derived AcrVA4 exploits specific recognition of the pre-crRNA processing nuclease by effectively mimicking the pre-crRNA substrate with the contribution of the carboxylates E184 and E204.

Our structure revealed that the CBD of AcrVA4 positions E184 to hydrogen bond with the 5'-hydroxyl on a mature crRNA, termini chemistry generated by Cas12a-catalyzed pre-crRNA cleavage (*Figure 2B*). Given this observation, we wondered if AcrVA4 would be able to block dsDNA *cis*-cleavage by LbCas12a in the presence of an unprocessed pre-crRNA substrate. To test this, we complexed LbCas12a with crRNA bearing either a 5'-$PO_4$ or 5'-dT(−21), RNA substrates that are uncleavable by Cas12a but mimic the structure of an unprocessed pre-crRNA. LbCas12a pre-complexed with a crRNA bearing a 5'-hydroxyl, mimicking the processed mature crRNA, was capable of robust dsDNA *cis*-cleavage that was sensitive to inhibition by AcrVA4 (*Figure 2E*). LbCas12a pre-complexed with crRNA bearing a 5'-$PO_4$ or 5'-dT(−21) were still effective at catalyzing dsDNA *cis*-cleavage; however, AcrVA4-mediated inhibition was compromised in the presence of a 5'-$PO_4$ crRNA and lost in the presence of an unprocessable 5'-dT(−21) pre-crRNA (*Figure 2E*). To demonstrate that the loss of AcrVA4 inhibition activity was due to a binding defect, we carried out size exclusion chromatography (SEC) experiments to assay binding between LbCas12a-crRNA and AcrVA4. Binding of AcrVA4 to the LbCas12a-crRNA complex was substantially decreased in the presence of crRNA bearing either a 5'-$PO_4$ or 5'-dT(−21), indicative of a binding defect (*Figure 2—figure supplement 2B*). Collectively, these structural and biochemical data demonstrate how the bacteriophage-derived AcrVA4 exploits recognition of the conserved Cas12a crRNA processing nuclease through mimicry of the pre-crRNA substrate to bind at the active site and contact the 5'-hydroxyl of the mature crRNA.

## AcrVA4 forms a heterotetrameric complex with LbCas12a-crRNA

Under our experimental conditions, the AcrVA4 homodimer assembles with LbCas12a-crRNA into a distribution of two complexes at equilibrium: a monomeric LbCas12a-crRNA bound to the AcrVA4 homodimer and a heterotetrameric assembly of two LbCas12a-crRNA complexes bound to an AcrVA4 homodimer (*Knott et al., 2019*). To investigate the nature of the heterotetrameric assembly and what effect it might have on the disposition of the bound Cas12a-crRNA complexes, we reprocessed our cryo-EM data to generate a map corresponding to the heterotetrameric arrangement. Taking a subset of 79,786 particle images identified by 3D classification, we reconstructed a 3D cryo-EM density map at 4.9 Å resolution representing State II (*Table 1*, *Figure 1—figure supplement 1*). Using this reconstructed map, two copies of our high-resolution State I LbCas12a-crRNA-AcrVA4 structure were rigid-body modeled to generate a butterfly shaped dimer of dimers (*Figure 3*). Each copy of the LbCas12a-crRNA complex is held against a CBD of AcrVA4, with $\alpha_1$ of each AcrVA4 protomer forming the heterotetramer interface (*Figure 3A*). Assessing the local environment surrounding the AcrVA4 dimer revealed that the two Cas12a molecules do not make any direct contacts to each other (*Figure 3B*), suggesting that the interaction between Cas12a and AcrVA4 is due solely to the CBD contact interface. While lower in resolution, this map of the heterotetrameric complex included additional density not visible in our higher resolution State I reconstruction, enabling further de novo modeling to extend the CBD $\alpha_1$ helix as a poly-alanine sequence (*Figure 3A*). Although EM density for the unmodelled N-terminal domain (residues 1–125) in each AcrVA4 protomer was clearly present (*Figure 3A*), it was not possible to model this region of the AcrVA4 dimer due to the limited resolution. Intriguingly, the $\alpha_1$ coiled-coil domain projects away from the interface with LbCas12a-crRNA and makes no additional contacts with either Cas12a protomer. This analysis of the heterotetramer architecture suggests that, although visually striking, the contacts stabilizing

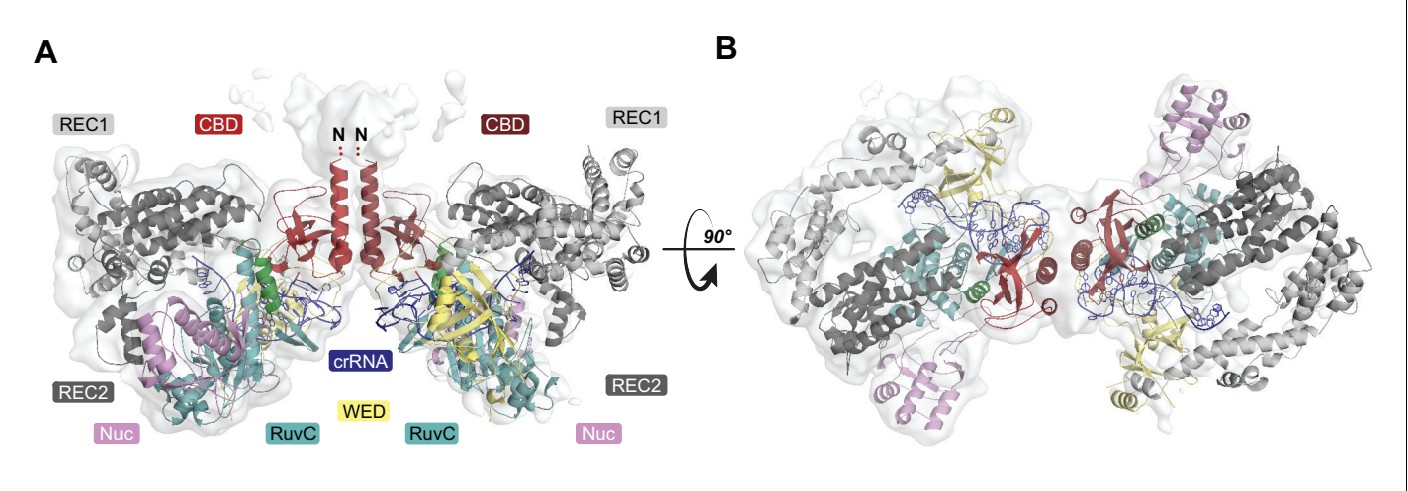

**Figure 3.** Overall structure of the heterotetrameric LbCas12a-crRNA-AcrVA4 complex. (A–B) Dimeric LbCas12a-crRNA complex (cartoon) bound to homodimeric AcrVA4 (cartoon, two shades of red) shown related by a 90° rotation.
DOI: https://doi.org/10.7554/eLife.49110.010

the interaction between AcrVA4 and Cas12a are limited to those observed and validated through inspection of our State I structure.

## AcrVA4 is an allosteric inhibitor of Cas12a DNA binding

AcrVA4 blocks dsDNA binding to the LbCas12a-crRNA complex (*Knott et al., 2019*). To recognize dsDNA, Cas12a binds a TTTV PAM sequence via the PAM-interacting (PI) domain, triggering crRNA strand invasion and base-pairing with the target strand (*Nishimasu et al., 2017*; *Swarts et al., 2017*; *Yamano et al., 2016*; *Nishimasu et al., 2017*). Our cryo-EM structures revealed that AcrVA4 associated at the Cas12a WED domain, far from the site of dsDNA recognition. This structural observation suggested that AcrVA4 might function through a mode of allosteric, rather than competitive, inhibition. To identify the nature of AcrVA4 allosteric control, we superimposed the LbCas12a-crRNA complex (PDB code: 5ID6) on our AcrVA4-bound state revealing that inhibitor binding did not change the overall architecture of the complex (r.m.s.d = 1.17 Å, *Figure 4—figure supplement 1A*). Careful inspection of the superimposed states revealed a set of subtle distortions to the LbCas12a bridge-helix in the AcrVA4 bound state (*Figure 4—figure supplement 1B*). To form a stable R-loop structure upon DNA binding, LbCas12a must undergo a large conformational change (*Figure 4—video 1*). Significant among these domain motions is the movement of the LbCas12a bridge-helix which drives a conserved arginine residue, R887, towards the cRNA: DNA heteroduplex (*Figure 4—video 2*) (*Yamano et al., 2016*; *Yamano et al., 2017*). In our structures, the bridge-helix is braced against AcrVA4 (*Figure 4A*) where AcrVA1 W178 is stacked against the conserved arginine (LbCas12a: R887) with its indole amino proton capping the bridge-helix main chain carbonyl oxygen of Cas12a E885 (*Figure 4B*). The architecture of this interaction suggested that AcrVA4 might use W178 to allosterically lock the bridge-helix to prevent the propagation of conformational changes required to stabilize R-loop formation (*Figure 4—video 3*) (*Swarts et al., 2017*; *Stella et al., 2017*; *Stella et al., 2018*).

To explore the role of W178 we generated AcrVA4-W178A and observed that this substition almost completely restored LbCas12a-mediated dsDNA *cis*-cleavage (*Figure 4C*). To verify that the loss of inhibition by AcrVA4-W178A was not due to misfolding, we assayed the ability of AcrVA4-W178A to form a complex with LbCas12a-crRNA. Using size-exclusion chromatography coupled to di-angle light scattering (SEC-DALS) we determined that, like wild-type AcrVA4, AcrVA4-W178A was itself a dimer (*Figure 4—figure supplement 2A–B*) able to form a monomeric or heterotetrameric assembly (*Figure 4D*, *Figure 4—figure supplement 2C–E*). Collectively, these data support a model of allosteric inhibition by AcrVA4 where conformational locking of the LbCas12a bridge-helix prevents the dynamics required for R-loop formation.

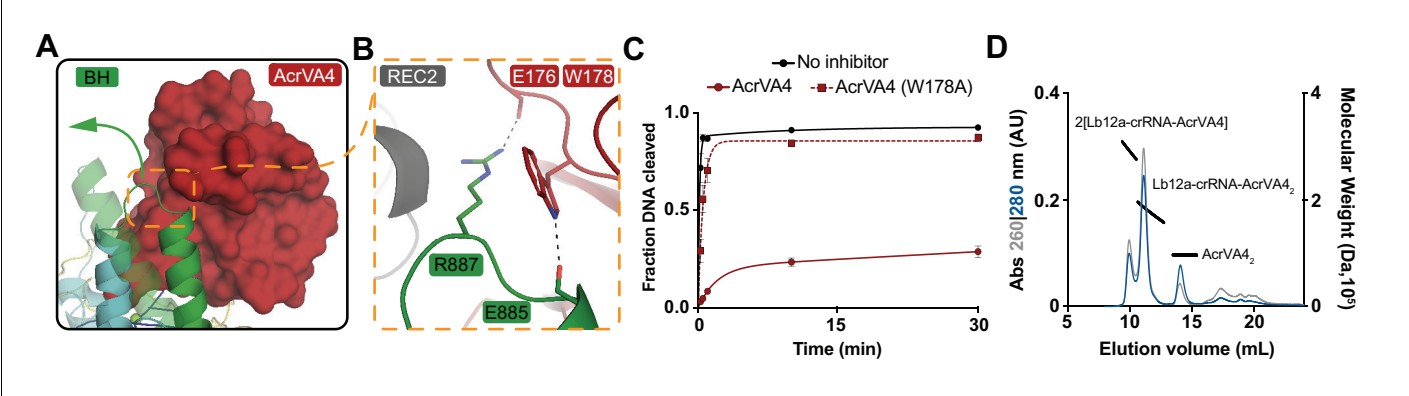

**Figure 4.** AcrVA4 locks the Cas12a bridge-helix to prevent DNA binding. (**A**) AcrVA4 (surface) is shown braced against the bridge-helix (BH, green, cartoon) near the RuvC (teal, cartoon). The conformation of the LbCas12a bridge-helix when bound to DNA is shown semi-transparent with a green arrow denoting the direction of helix motion upon DNA binding. (**B**) Detailed atomistic view of AcrVA4 (red) recognition of the Cas12a bridge-helix (green). (**C**) LbCas12a dsDNA *cis*-cleavage over time measured under single-turnover conditions in the presence or absence of AcrVA4 containing alanine substitutions (mean ∓ s.d., *n* = 3 independent measurements). Two-phase exponential decay experimental fits are shown as solid or dashed lines. (**D**) Size-exclusion chromatography coupled di-angle light scattering (SEC-DALS) trace for AcrVA4 W178A in the presence of LbCas12a-crRNA. The absorbance at 280 nm (blue) and 260 nm (gray) are shown (left axis) with the linear region for the mass estimate corresponding to the relevant peaks (black lines, right axis).

DOI: https://doi.org/10.7554/eLife.49110.011

The following video and figure supplements are available for figure 4:

**Figure supplement 1.** Superposition of LbCas12a-crRNA (PDB code: 5ID6) on our State I LbCas12a-crRNA-AcrVA4 complex.

DOI: https://doi.org/10.7554/eLife.49110.012

**Figure supplement 2.** Size exclusion chromatography coupled di-angle light scattering (SEC-DALS) for AcrVA4 and LbCas12a-crRNA.

DOI: https://doi.org/10.7554/eLife.49110.013

**Figure 4—video 1.** LbCas12a undergoes a large conformational change from the crRNA-bound state to a DNA-bound state.

DOI: https://doi.org/10.7554/eLife.49110.014

**Figure 4—video 2.** The LbCas12a bridge-helix undergoes a large conformational change upon DNA binding.

DOI: https://doi.org/10.7554/eLife.49110.015

**Figure 4—video 3.** AcrVA4 binding prevents LbCas12a bridge-helix conformational dynamics upon DNA binding.

DOI: https://doi.org/10.7554/eLife.49110.016

## The C-terminal binding domain of AcrVA4 is sufficient to inhibit Cas12a

Our cryo-EM structures and accompanying biochemistry revealed that the CBD of AcrVA4 binds the pre-crRNA processing nuclease and allosterically gates key Cas12a conformational changes to inhibit DNA binding. Intriguingly, both wild-type and AcrVA4-W178A form obligate homodimers that associate with one or two copies of LbCas12a-crRNA (*Figure 4D*, *Figure 4—figure supplement 2*). However, LbCas12a-crRNA complexes formed with AcrVA4-W178A can catalyze dsDNA *cis*-cleavage (*Figure 4C*). These data suggested that higher order assembly alone was insufficient to block Cas12a dsDNA targeting. To test this hypothesis, we generated a dimerization incompetent AcrVA4 truncation (AcrVA4 Δ1–134) bearing only the CBD as resolved in our higher resolution cryo-EM structure (*Figure 1D*). We reasoned that this construct would exist as a monomer in solution and bind with 1:1 stoichiometry to an LbCas12a-crRNA complex. To determine the solution behavior of AcrVA4 Δ1–134, we carried out size-exclusion chromatography coupled to small-angle X-ray scattering (SEC-SAXS) experiments (*Figure 5A–B*). Using SEC-SAXS analysis, we observed that the full-length AcrVA4 was considerably larger than the truncated AcrVA4 Δ1–134 as indicated by its greater radius of gyration ($R_g$), maximum dimension ($D_{max}$), and calculated MALS and SAXS molecular weights (*Table 2*, *Figure 5—figure supplement 1A*). Furthermore, normalized Kratky analysis suggested that each construct was well ordered (*Figure 5—figure supplement 1C*). Using atomistic models (*Figure 5A*), we generated theoretical SAXS profiles which accurately fit the experimental SAXS data, suggesting that full-length AcrVA4 was dimeric while AcrVA4 Δ1–134 was monomeric in solution (*Figure 5B* and *Table 2*). To assay the stoichiometry of AcrVA4 Δ1–134 binding to

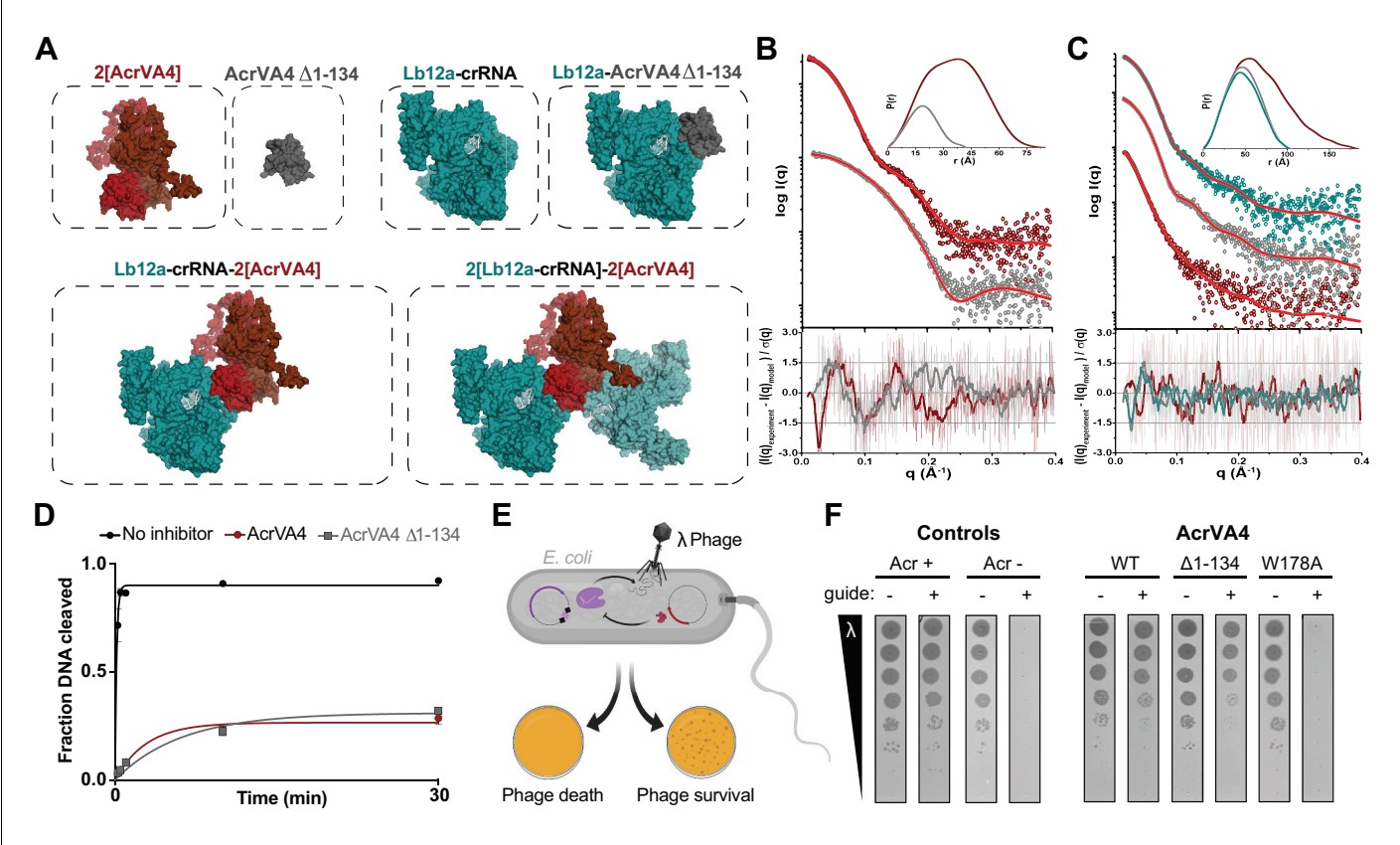

**Figure 5.** The C-terminal binding domain is sufficient for Cas12a inhibition. (**A**) Atomistic models for the AcrVA4, AcrVA4 Δ1–134, LbCas12a-crRNA, LbCas12a-crRNA-AcrVA4 Δ1–134 and LbCas12a-crRNA-AcrVA4 that were used to match experimental small-angle X-ray scattering (SAXS) data shown in panel B and panel C. (**B**) Experimental data for AcrVA4, AcrVA4 Δ1–134 (red and gray) and theoretical (red line) SAXS profiles for the solution state models shown in the panel A. SAXS fits are shown together with the fit-residuals. (**C**) Experimental data for LbCas12a-crRNA, LbCas12a-crRNA-AcrVA4 Δ1–134 and LbCas12a-crRNA-AcrVA4 (teal, gray and red) and theoretical (red line) SAXS profiles for the solution state models shown in the panel A. SAXS fits are shown together with the fit-residuals. (B/C-insets) Normalized $P(r)$ determined from the experimental SAXS curves. The area of each $P(r)$ is normalized relative to the SAXS calculated molecular weights (**Table 2**). (**D**) LbCas12a dsDNA *cis*-cleavage over time measured under single-turnover conditions in the presence or absence of AcrVA4 or AcrVA4 Δ1–134 (mean ∓ s.d., $n$ = 3 independent measurements). Two-phase exponential decay experimental fits are shown as solid lines. (**E**) Schematic representation of phage lambda plaque assay in *E. coli*. All strains harbor both a CRISPR-Cas plasmid (purple) and anti-CRISPR plasmid (red). Cas12a confers immunity to phage lambda, while anti-CRISPR inhibition of Cas12a restores plaquing. (**F**) Phage plaque assay to compare inhibition of LbCas12a by wild-type AcrVA4 and mutants relative to positive (AcrVA1: Acr +) and negative (AcrIIA4: Acr -) control anti-CRISPRs in *E. coli*. Ten-fold serial dilutions of heat-inducible phage lambda spotted on lawns of *E. coli* strains expressing the specified anti-CRISPR protein and a non-targeting guide (-) or lambda-targeting guide (+). Images shown are representative of the effect seen in replicates ($n$ = 3 independent measurements).

DOI: https://doi.org/10.7554/eLife.49110.017

The following figure supplements are available for figure 5:

**Figure supplement 1.** Small-angle X-ray scattering (SAXS) and multi-angle light scattering (MALS) data.
DOI: https://doi.org/10.7554/eLife.49110.018
**Figure supplement 2.** Phage lambda plaque assays.
DOI: https://doi.org/10.7554/eLife.49110.019

LbCas12a-crRNA, we collected SEC-SAXS data for the uninhibited complex, the AcrVA4 bound complex, and the AcrVA4 Δ1–134 bound complex. SEC-SAXS analysis revealed that the solution scattering of the LbCas12a-crRNA complex agreed well with the published X-ray crystal structure (**Table 2**, **Figure 5C**, **Figure 5—figure supplement 1B and D**). Analysis of the solution scattering from LbCas12a-crRNA-AcrVA4 revealed a mixture of monomeric and heterotetrameric arrangements, consistent with our structures describing States I and II (**Table 2**, **Figure 5C**, **Figure 5—figure supplement 1B and D**). Finally, SEC-SAXS analysis for LbCas12a-crRNA in the presence of AcrVA4 Δ1–

**Table 2.** SEC-SAXS-MALS-UV-vis data.

| Sample | Theoretical Mwt (kDa) | SAXS Mwt (kDa) | MALS Mwt (kDa) | Rg (Å) | Dmax (Å) | Px | Fit χ2 |
|---|---|---|---|---|---|---|---|
| AcrVA4 | 55.2 | ~55 | 55.9 (±0.092%) | 27.71 (±0.27) | 85 | 4 | 1.68 |
| AcrVA4 (Δ1–134) | 12.1 | ~12 | 12.6 (±0.628%) | 14.72 (±0.16) | 42 | 4 | 1.71 |
| Lb12a-crRNA | 156 | ~140 | 160.2 (±0.041%) | 36.15 (±0.46) | 103 | 4 | 1.39 |
| Lb12a-crRNA-AcrVA4 | 212 (369) | ~265 | 415.1 (±0.012%) | 52.76 (±2.55) | 183 | 3.9 | 1.75 |
| Lb12a-crRNA-AcrVA4 (Δ1–134) | 169 | ~155 | 176.5 (±0.044%) | 36.90 (±0.42) | 105 | 4 | 1.31 |

DOI: https://doi.org/10.7554/eLife.49110.020

134 revealed a subtle but significant increase in particle size (*Figure 5C*) and calculated mass (*Table 2*), reflecting a single bound AcrVA4 Δ1–134. Furthermore, an atomistic model of a 1:1 LbCas12a-crRNA-AcrVA4 Δ1–134 complex accurately described the experimental scattering data (*Table 2*, *Figure 5C*). Collectively, these results indicated that AcrVA4 Δ1–134 was monomeric and formed complexes with LbCas12a-crRNA that have 1:1 stoichiometry. This stoichiometric arrangement contrasts with the apparently obligatory dimerization of full-length AcrVA4 and its monomeric-heterotetrameric equilibrium with LbCas12a-crRNA.

We next tested whether AcrVA4 Δ1–134 could prevent dsDNA *cis*-cleavage by pre-incubating LbCas12a-crRNA with inhibitor prior to the addition of radiolabeled dsDNA substrate. We observed that AcrVA4 Δ1–134 alone was sufficient to inhibit dsDNA *cis*-cleavage by LbCas12-crRNA (*Figure 5D*). To directly assess if this truncation of AcrVA4 was capable of countering Cas12a-mediated immunity, we assayed anti-CRISPR restoration of phage activity during infection in an *E. coli* phage lambda plaque assay (*Figure 5E*). Whereas LbCas12a-crRNA targeting the phage genome confers immunity from lambda infection in *E. coli*, expression of either AcrVA4 or AcrVA4 CBD efficiently inhibits LbCas12a and restores plaquing by the targeted phage to levels comparable to that observed in the absence of a targeting spacer (*Figure 5F*, *Figure 5—figure supplement 2*). Specifically, no difference in efficiency of plaquing between full-length and truncated AcrVA4 was observed, although cells expressing the truncated variant formed smaller plaques (*Figure 5F*, *Figure 5—figure supplement 2*). Collectively, our in vitro biochemical and in vivo plaque assays indicated that the AcrVA4 CBD was sufficient for inhibition of LbCas12a-mediated phage interference. Furthermore, these data underscore that the N-terminal domain of AcrVA4 is apparently dispensable for inhibition of LbCas12a both in vitro and in vivo.

## AsCas12a evades AcrVA4 by concealing the pre-crRNA processing nuclease

In contrast to the broad spectrum inhibition mediated by AcrVA1 (*Knott et al., 2019*; *Watters et al., 2018*), AcrVA4 is selective for specific Cas12a enzymes and has no effect on the Cas12a ortholog AsCas12a. Due to the low sequence identity between Cas12a orthologs (~30%), it was not possible to attribute any single difference at the sequence level to AcrVA4 susceptibility. Careful inspection of the LbCas12a-crRNA-AcrVA4 binding interface superimposed onto the available AsCas12a ternary structure revealed the presence of a helical bundle in the AsCas12a WED domain that forms a compact arrangement proximal to the pre-crRNA processing nuclease (*Figure 6A*, *Figure 6—figure supplement 1*). In contrast, the same region in LbCas12a forms a short hairpin-turn above the pre-crRNA processing nuclease, leading back into the core of the WED domain (*Figure 2A*, *Figure 6—figure supplement 1*). These structural observations suggested that AcrVA4 might be sterically occluded from binding to the AsCas12a WED domain due to the helical bundle. Consistent with this, superimposing the AcrVA4 bound state of LbCas12a-crRNA onto the AsCas12a ternary structure revealed that the WED domain helical bundle would sterically clash with AcrVA4 (*Figure 6A*).

To determine if the helical bundle governs AsCas12a resistance to AcrVA4, we generated reciprocal swaps of the helical bundle between LbCas12a and AsCas12a to create Cas12a chimeras (*Figure 6B*). Both chimeric enzymes (As*Cas12a and Lb*Cas12a) exhibited modest (~five fold) reductions in the rate of pre-crRNA processing (*Figure 6—figure supplement 2A and B*) but maintained near wild-type dsDNA *cis*-cleavage activity in the absence of AcrVA4 (*Figure 6C*, *Figure 6—figure*

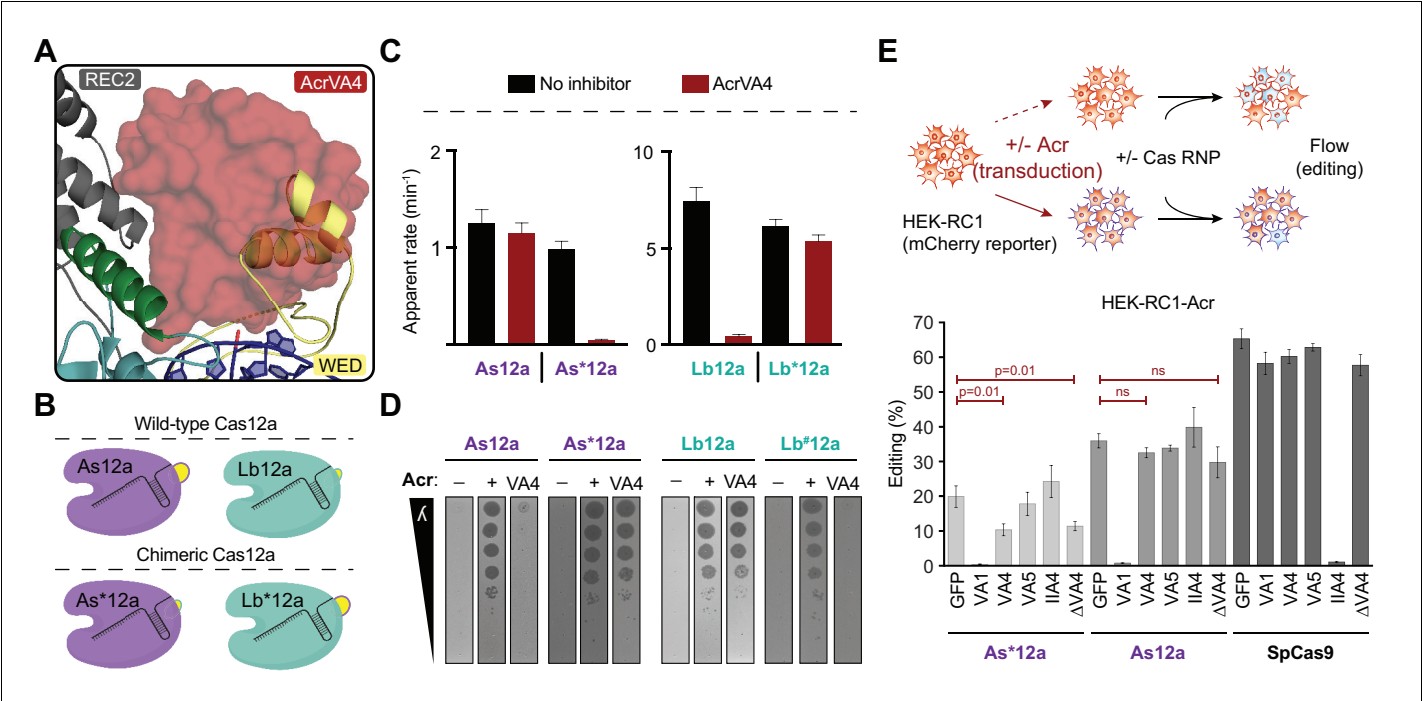

**Figure 6.** AsCas12a evades AcrVA4 by concealing its pre-crRNA processing nuclease. (A) The C-terminal binding domain of AcrVA4 (surface) is shown superposed on the AsCas12a structure where it clashes with the WED domain (yellow, cartoon). (B) Schematic representation of the wild-type (Lb and As) and engineered chimeric (Lb* and As*) Cas12a constructs. (C) Bar-graph illustrating the apparent rate of wild-type or chimeric Cas12a-mediated dsDNA *cis*-cleavage under single-turnover conditions in the presence or absence of AcrVA4 (mean ∓ s.d., *n* = 3 independent measurements). Single-phase exponential fits to the cleavage kinetics from which the rate was derived can be found in *Figure 6—figure supplement 2C*. (D) Phage plaque assays to determine susceptibility of chimeric Cas12a to AcrVA4 in *E. coli*. Ten-fold serial dilutions of heat-inducible phage lambda spotted on lawns of *E. coli* strains expressing lambda-targeting guide, wild-type or chimeric (*/#) Cas12a, and AcrVA4 or the indicated control anti-CRISPR protein. Images shown are representative of the effect seen in replicates (*n* = 3 independent measurements). (E) CRISPR-Cas12a inhibition specificity in human cells. Schematic (top panel) showing human cells stably expressing a fluorescence reporter and doxycycline-inducible anti-CRISPR (Acr) constructs. Acr expression blocks genome editing upon transfection of susceptible Cas ribonucleoprotein (RNP) complexes, quantifiable by flow cytometry of mCherry fluorescence. Assessment of editing efficiency in HEK-RC1 reporter cells (bottom panel) expressing GFP or GFP-Acr polycistronic constructs (AcrVA1, AcrVA4, AcrVA5, AcrIIA4, AcrVA4 Δ1–134) and transfected with various Cas9 and Cas12a RNPs targeting the reporter. Note, in contrast to wild-type AsCas12a (As12a), editing by the AsCas12a-chimera (As*12a) was moderately susceptible to AcrVA4 and AcrVA4 Δ1–134 inhibition.

DOI: https://doi.org/10.7554/eLife.49110.021

The following figure supplements are available for figure 6:

**Figure supplement 1.** Multiple sequence alignment of Cas12a orthologs.
DOI: https://doi.org/10.7554/eLife.49110.022

**Figure supplement 2.** Wild-type and chimeric Cas12a pre-crRNA processing and dsDNA cleavage.
DOI: https://doi.org/10.7554/eLife.49110.023

**Figure supplement 3.** Optimizing induction of Cas12a chimeras in phage lambda plaque assay.
DOI: https://doi.org/10.7554/eLife.49110.024

**Figure supplement 4.** Human CRISPR-Cas and anti-CRISPR (Acr) genome editing assay.
DOI: https://doi.org/10.7554/eLife.49110.025

**Figure supplement 5.** Phylogenetic reconstruction for Cas12a orthologs.
DOI: https://doi.org/10.7554/eLife.49110.026

*supplement 2C*). In the presence of AcrVA4, wild-type AsCas12a evades AcrVA4 whereas wild-type LbCas12a is robustly inhibited (*Figure 6C*). In contrast, the chimeras exhibited a complete phenotype swap where As*Cas12a was robustly inhibited by AcrVA4 while Lb*Cas12a maintained a near wild-type level of dsDNA *cis*-cleavage (*Figure 6C*). We next investigated if the exposed vulnerability to AcrVA4 in engineered As*Cas12a would support anti-CRISPR rescue of phage lambda infection. While the chimeric effector maintained the capacity to confer immunity to phage lambda, removal of the helical bundle rendered the enzyme sensitive to AcrVA4, efficiently restoring plaquing of

targeted phage to levels consistent with the positive control anti-CRISPR protein (*Figure 6D*, *Figure 5—figure supplement 2*, *Figure 6—figure supplement 3*). In contrast, the Lb*Cas12a chimera was sensitive to AcrVA4 inhibition during phage lambda infection, despite efficient phage interference in the absence of Type V-A anti-CRISPRs (*Figure 5—figure supplement 2*, *Figure 6—figure supplement 3*). This result suggested that the AsCas12a helical bundle might somehow destabilize the Lb*Cas12a chimera in *E. coli*. To circumvent this, we created an alternative chimera, Lb#Cas12a, by inserting a putative helical bundle from a closely related *L. bacterium* strain (OF09-6) (*Figure 6—figure supplement 1*). In contrast to Lb*Cas12a, Lb#Cas12a achieved complete immunity against phage lambda, even in the presence of AcrVA4 (*Figure 6C*).

Given that AcrVA4-susceptible As*Cas12a maintained DNA targeting activity, we wondered if it might be susceptible to AcrVA4 in the context of mammalian genome editing. To test this, we generated C-terminal NLS-tagged constructs of the wild-type AsCas12a and the AcrVA4 sensitive chimera for ribonucleoprotein delivery into HEK293T-based human reporter cells stably expressing mCherry and one of several anti-CRISPRs (*Figure 6E*, *Figure 6—figure supplement 4*). Consistent with our biochemical data, wild-type AsCas12a was not susceptible to AcrVA4 whereas the chimeric enzyme was modestly inhibited from inducing genome edits in the presence of either AcrVA4 or the C-terminal truncated form AcrVA4 Δ1–134 (*Figure 6E*). Taken together, these data suggest that the limited spectrum of inhibition available to some anti-CRISPRs might be expanded by protein engineering, effectively reprogramming anti-CRISPR susceptibilities.

Our experimental exchange of the two-helix bundle between LbCas12a and AsCas12a suggested that this structural feature might represent an evolutionary path to escape AcrVA4-like activity through WED domain insertion. To explore this possibility, we generated a phylogenetic reconstruction of Cas12a orthologs rooted to the proposed ancestral transposon-encoded nuclease TnpB (*Shmakov et al., 2015*). We found that Cas12a orthologs harboring the helical bundle appeared more closely related to TnpB than Cas12a variants lacking the bundle (*Figure 6—figure supplement 5A*). This was further supported by a phylogenetic tree constructed from the RuvC domains of Cas12a and TnpB, an inference independent from the presence or absence of the helical bundle (*Figure 6—figure supplement 5B*). These observations suggest that AsCas12a containing the helical bundle is likely more closely related to a Cas12a common ancestor than LbCas12a which lacks the helical bundle. In summary, these results reveal an apparent evolutionary trajectory describing the loss of an ancestral two-helix bundle which may have driven opportunistic co-evolution of the AcrVA4 inhibitor by bacteriophage.

## Discussion

Anti-CRISPRs (Acr) have co-evolved with CRISPR-Cas proteins to provide bacteriophage with broad-spectrum, or in some cases, highly selective protection from RNA-guided destruction (*Pawluk et al., 2018*; *Hwang and Maxwell, 2019*). Inhibitors of Cas12a, AcrVA1 and AcrVA5, each utilize a distinct mechanism to enzymatically inhibit DNA targeting (*Suresh et al., 2019*; *Knott et al., 2019*; *Dong et al., 2019*). Here, we present structural and functional data demonstrating that the inhibition mechanism of AcrVA4 is also unique.

CRISPR-Cas12a, like Cas13a, are systems that carry out autonomous CRISPR array processing (*Zetsche et al., 2015*; *Fonfara et al., 2016*; *East-Seletsky et al., 2016*). We demonstrated that AcrVA4 specifically recognizes Cas12a by binding directly to this pre-crRNA processing nuclease and the mature crRNA 5'-end. While a number of Acrs have been described that effectively mimic target DNA to block DNA binding, AcrVA4 is the first example of an anti-CRISPR that exploits recognition of a pre-crRNA processing nuclease. Binding at the site of pre-crRNA processing positions AcrVA4 to allosterically lock the bridge-helix of Cas12a to prevent dsDNA target recognition and subsequent interference, consistent with a recent study (*Zhang et al., 2019*). The existence of such bacteriophage-derived inhibitors targeting the pre-crRNA processing nuclease highlights a unique vulnerability in CRISPR-Cas adaptive immunity. Other single component effectors that directly catalyze pre-crRNA processing, including Cas13, might also be susceptible to a similar mode of inhibition, although such Acrs have not yet been identified. It also raises the possibility that bacteriophage might evolve inhibitors targeted to alternative pre-crRNA processing machinery such as Csy4 (*Przybilski et al., 2011*) or Cas6 (*Carte et al., 2008*). Finally, the specific recognition of the crRNA

5'-end chemistry by AcrVA4 provides opportunities to create orthogonal enzymes bearing modified crRNA that do or do not respond to AcrVA4 in genome editing applications.

Unlike other Acrs that prevent DNA binding by Cas9 or Cas12a, AcrVA4 is unusually large (234 amino acids) and forms a dimer whose binding to Cas12a does not competitively inhibit the DNA binding site in Cas12a. Although dimerization of AcrVA4 and formation of a heterotetrameric interaction with Cas12a is structurally striking, our biochemical data, phage interference assays, and human gene editing data demonstrated that dimerization is not necessary for inhibition. However, we did observe subtle phenotypic differences between AcrVA4 and AcrVA4 Δ1–134, observations that could be attributed to distinct rates of protein translation and turnover which can vary independently of transcriptional inducer concentration. It is possible that the N-terminal domain of AcrVA4 serves an additional function in an endogenous context to provide a selective advantage against activated and *trans*-cleaving Cas12a. For example, the dimerization of Cas12a may elicit a cellular response influencing Cas12a half-life or cellular localization. However, further experiments are needed to elucidate the true function of the apparently dispensable N-terminal domain of AcrVA4.

Our data also suggest that structural variability in divergent Cas12a effectors may be a driver for opportunistic co-evolution of specific bacteriophage inhibitors. The narrow spectrum of inhibition for AcrVA4 results from the presence of a structural feature in some Cas12a orthologs that occludes inhibitor binding. Although such a structural elaboration might have been expected to evolve as an insertion to protect Cas12a from Acr targeting, phylogenetic analysis suggested that this AcrVA4 shield is instead ancestral and was lost in a group of Cas12a orthologs after divergence from a common ancestor (*Figure 7*). These data suggest that the fitness advantage driving helical-bundle deletion created the co-evolutionary opportunity by exposing an exploitable site on Cas12a. This raises the possibility that anti-CRISPR surveillance may drive selection in bacteria for Cas enzymes bearing local insertions or deletions that have little to no effect on the activity of functionally constrained elements such as the pre-crRNA processing nuclease or RuvC nuclease. In this way, bacteria might avoid anti-CRISPR activity by accommodating indels that do not interfere with activity but provide steric protection from Acr association. Such a process could help explain the prevalence of large indels seemingly scattered at random throughout CRISPR-Cas effectors and the diversification of CRISPR systems leading to anti-CRISPR resistence (*Pausch et al., 2017*). Understanding the modes

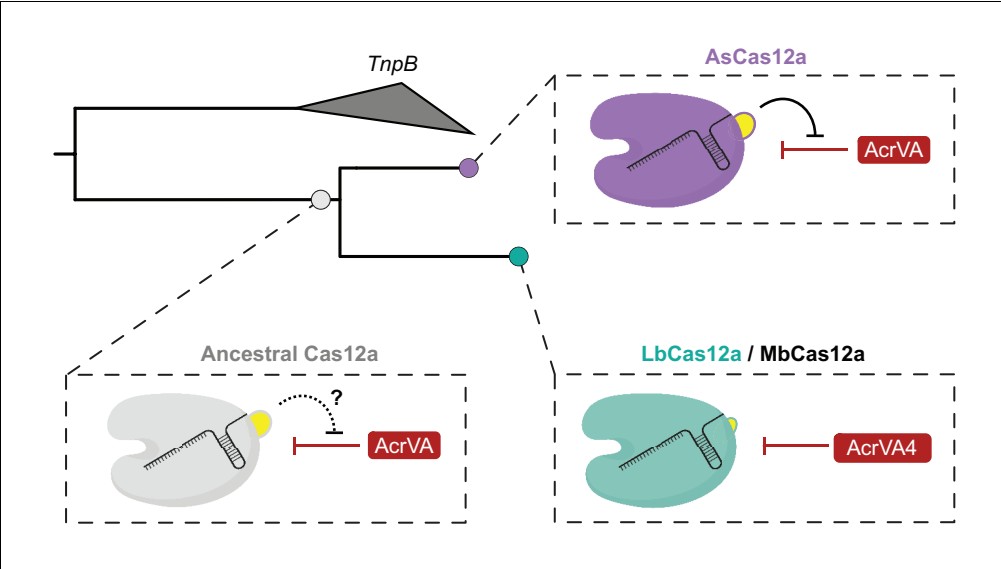

**Figure 7.** Model for the co-evolution of AcrVA4 susceptibility. The ancestral Cas12a likely possessed a helical bundle, hindering any exploitation of the processing nuclease and bridge-helix (bottom left). While AsCas12a maintained the ancestral helical bundle and resistance to Acr activity at that site (top right), LbCas12a and related orthologs lost the helical bundle, providing a co-evolutionary opportunity for AcrVA4 allosteric inhibition.
DOI: https://doi.org/10.7554/eLife.49110.027

of anti-CRISPR inhibition will continue to reveal new aspects of CRISPR biology and provide opportunities to harness these inhibitors to control Cas activities in biotechnological applications.

# Materials and methods

## Key resources table

| Reagent type (species) or resource | Designation | Source or reference | Identifiers | Additional information |
|---|---|---|---|---|
| Peptide, recombinant protein | *AcrVA4* | *Watters et al., 2018* | Addgene #115656 | |
| Peptide, recombinant protein | *LbCas12a* | *Watters et al., 2018* | Addgene #115656 | |
| Peptide, recombinant protein | *AsCas12a* | *Watters et al., 2018* | Addgene #113430 | |
| Strain, strain background (*E. coli*) | NEB 10-beta | New England Biolabs | | |
| Strain, strain background (*E. coli*) | MG1655 λ+ (cl857 bor::kanR) | DOI: 10.1073/pnas.0808831105 | | |
| Strain, strain background (*E. coli*) | Rosetta 2 (DE3) | Novagen | | |
| Strain, strain background (*lambda*) | *λ phage* | This paper | | cl857 bor::kanR |
| Sequence-based reagent | LbCas12a pre-crRNA | IDT | lab archive: rGJK_006 | rArGrArUrUrArArArUrArArUrUrUrCrUrArCrUrArArGrUrGrUrArGrArUrArArArGrUrGrCrUrCrArUrCrArUrUrGrGrArArArArCrGrU |
| Sequence-based reagent | AsCas12a pre-crRNA | IDT | lab archive: rGJK_008 | rGrArCrUrUrUrUrArUrUrUrCrUrArCrUrCrUrUrGrUrArGrArUrArArArArGrUrGrCrUrCrArUrCrArUrUrGrGrArArArArCrGrU |
| Sequence-based reagent | LbCas12a crRNA | IDT | lab archive: rGJK_017 | rArArUrUrUrCrUrArCrUrArArGrUrGrUrArGrArUrGrArUrCrGrUrUrArCrGrCrUrArArCrUrArUrGrA |
| Sequence-based reagent | AsCas12a crRNA | IDT | lab archive: rGJK_018 | rArArUrUrUrCrUrArCrUrCrUrUrGrUrArGrArUrGrArUrCrGrUrUrArCrGrCrUrArArCrUrArUrGrA |
| Sequence-based reagent | Non-target DNA strand | IDT | lab archive: dGJK_006 | GAC GAC AAA ACT TTA GAT CGT TAC GCT AAC TAT GAG GGC TGT CTG TGG AAT GCT A |
| Sequence-based reagent | Target DNA strand | IDT | lab archive: dGJK_007 | TAG CAT TCC ACA GAC AGC CCT CAT AGT TAG CGT AAC GAT CTA AAG TTT TGT CGT C |

*Continued on next page*

*Continued*

| Reagent type (species) or resource | Designation | Source or reference | Identifiers | Additional information |
|---|---|---|---|---|
| Sequence-based reagent | LbCas12a crRNA (dT −21) | IDT | lab archive: rGJK_051 | rArGrArUrUrArArATrArAr UrUrUrCrUrArCrUrArArGr UrGrUrArGrArUrArArArGr UrGrCrUrCrArUrCrArUrUr GrGrArArArArCrGrU |
| Sequence-based reagent | LbCas12a crRNA (5′-PO$_4$) | IDT | lab archive: rGJK_053 | /5Phos/rArUrUrUrCrUrAr CrUrArArGrUrGrUrArGrAr UrArArArGrUrGrCrUrCrAr UrCrArUrUrGrGrArArArAr CrGrU |
| Software, algorithm | Prism7 | GraphPad | | |
| Software, algorithm | Image QuantTL | GE Healthcare | | |
| Software, algorithm | ScÅtter 3.0 | BioIsis | | |

## Protein expression and purification

Plasmids encoding *Lachnospiraceae bacterium* (ND2006) Cas12a (Addgene #113431), *Acidaminococcus sp.* (BV3L6) Cas12a (Addgene #113430), and AcrVA4 (Addgene #115656) were generated from a custom pET-based expression vector as described previously (*Watters et al., 2018*). Cas12a or AcrVA4 point mutations, truncations, and chimeras were generated by either around-the-horn PCR or Gibson Assembly verified by Sanger DNA sequencing. Proteins were purified as described previously (*Watters et al., 2018*; *Knott et al., 2019*). Briefly, *E. coli* Rosetta 2 (DE3) containing Cas12a or AcrVA expression plasmids were grown in Lysogeny Broth overnight with ampicillin (100 µg mL$^{-1}$). Overnight cultures were subcultured in Terrific Broth to an OD$_{600}$ of 0.6–0.8, after which the cultures were cooled on ice for 15 min before induction with 0.5 mM IPTG and incubated overnight at 16°C for 16 hr. Cells were harvested by centrifugation and resuspended in wash buffer (20 mM Tris-Cl, (pH 7.5), 500 mM NaCl, 1 mM TCEP, 5% (v/v) glycerol) supplemented with 0.5 mM PMSF and cOmplete protease inhibitor (Roche), lysed by sonication, and purified over Ni-NTA Superflow resin (Qiagen) in wash buffer supplemented with either 10 mM imidazole (wash) or 300 mM imidazole (elution). Eluted proteins were digested overnight with TEV protease at 4°C in a Slide-A-Lyzer (10 kDa MWCO, Thermofisher) against dialysis buffer (20 mM Tris-Cl (pH 7.5), 125 mM NaCl, 1 mM TCEP, 5% (v/v) glycerol). Digested proteins were loaded onto an MBP-Trap (GE Healthcare) upstream of a Heparin HiTrap (GE Healthcare, Cas12a) or a HiTrap Q (GE Healthcare, AcrVA4) and eluted over a salt gradient (20 mM Tris-Cl, (pH 7.5), 1 mM TCEP, 5% (v/v) glycerol, 125 mM – 1 M KCl). The eluted protein was concentrated before injection to a Superdex 75 or 200 10/300 Increase (GE Healthcare) developed in 20 mM HEPES-K (pH 7.5), 200 mM KCl, 1 mM TCEP, 5% (v/v) glycerol). Purified proteins were concentrated and snap frozen in LN$_2$ for storage at −80°C. The purity and integrity of proteins used in this study were assessed by SDS-PAGE (Coomassie blue staining) (*Supplementary file 1*).

## Electron microscopy sample preparation, data collection, and 3D reconstruction

LbCas12a complexes were prepared for cryo-EM in a buffer containing 20 mM HEPES-K (pH 7.5), 200 mM KCl, 1 mM TCEP, 1 mM MgCl$_2$ and 0.25% glycerol. Immediately after glow-discharging the grid for 14 s using a Solaris plasma cleaner, 3.6 µl droplets of the sample (~3 µM) were placed onto C-flat grids with 2 µm holes and 2 µm spacing between holes (Protochips). The grids were rapidly plunged into liquid ethane using a FEI Vitrobot MarkIV maintained at 8°C and 100% humidity, after being blotted for 4.5 s with a blot force of 8. Data were acquired using an FEI Titan Krios transmission electron microscope operated at 300 keV with a GIF energy filter, at a nominal magnification of 135,000X (0.9 Å pixel size), with defocus ranging from −0.7 to −2.1 µm. Micrographs were recorded using SerialEM (*Mastronarde, 2003*) on a Gatan K2 Summit direct electron detector operated in

super-resolution mode. We collected a 4.8 s exposure fractionated into 32, 150 ms frames with a dose of 9.6 $e^-\text{Å}^{-2}\text{s}^{-1}$. For single particle analysis, 33350 movies were collected. The 30 frames (the first two frames were skipped) of each image stack in super-resolution mode were aligned, decimated, summed and dose-weighted using Motioncor2 (*Zheng et al., 2017*). CTF values of the summed-micrographs were determined using Gctf (*Zhang, 2016*). Approximately 980,000 particles were picked by Gautomatch and then imported into CryoSparc (*Punjani et al., 2017*) for 2D analysis. Approximately 324,336 particles which contributed to good 2D class averages were selected for further *ab initio* reconstruction generating four classes. Class 2 (20.1% particles) and Class 3 (28.3% particles) displayed similar architecture and were combined for homogenous refinement producing a 3D cryo-EM reconstruction at the resolution of 2.99 Angstroms (State I). The particles in Class 3 (24.6%) displayed a dimeric architecture and were used for homogenous reconstruction of a 3D cryo-EM map at the resolution of 4.91 Angstroms (State II).

## Model building, refinement, and validation

For State I, the published structure of the LbCas12a-crRNA complex (PDB code: 5ID6) (*Dong et al., 2016*) was used as an input model after correcting the protein sequence, removing heteroatoms, and resetting temperature factors. The resulting model was refined against the final overall reconstruction at 2.99 Å resolution using the real space refinement program in PHENIX (*Afonine et al., 2018*; *Adams et al., 2010*) with validation using MOLPROBITY (*Chen et al., 2010*). Initially, the input model describing LbCas12a-crRNA was rigid body refined before simulated annealing and morphing with gradient-driven minimization. After manual inspection of the model in COOT (*Emsley et al., 2010*), we noted that the PI domain was largely disordered as were segments of the REC1/REC2 domains and these regions were removed from the model. Subsequent refinement was carried out using global minimization, morphing, local grid search, and temperature factor refinement with appropriate restraints (secondary structure, $C_B$, rotamer, and ramachandran restraints). The structure of residues 135–233 of AcrVA4 was then traced and built completely de novo based off the known amino acid sequence and subjected to iterative refinement as described above. For State II, the refined model from State I was duplicated and a C2 rotation applied centered on AcrVA4 $\alpha_1$ for rigid body refinement against the final overall reconstruction at 4.91 Å. The $\alpha_1$ helix of AcrVA4 was extended as an ideal poly-alanine $\alpha$-helix to from residues 126–134. Because of the overall lower resolution of the State II 3D reconstruction, the heterotetrameric model was also restrained by reference restraints generated from the completed model of State I.

## Figure generation and data deposition

Figures and conformational morphs were created using PyMol (The PyMol Molecular Graphics System, Version 1.8 Schrödinger, LLC). Molecular contacts and the total surface area buried in interface was determined by *jsPISA* (*Krissinel, 2015*). The cryo-EM map of the LbCas12a-crRNA-AcrVA4 complex at 2.99 Å resolution (State I) and the refined coordinates model have been deposited to the EMDB and PDB with accession codes EMD-20266 and PDB-6P7M, respectively. The cryo-EM map of the 2[LbCas12a-crRNA-AcrVA4] complex at 4.91 Å resolution (State II) and the refined coordinates have been deposited to the EMDB and PDB with accession codes EMDB-20267 and PDB-6P7N, respectively.

## RNA and DNA preparation

RNA used in this study were ordered from Integrated DNA Technologies (IDT). RNA substrates were purified by gel extraction from 12% (v/v) urea-denaturing PAGE (0.5X TBE) and ethanol precipitation. All DNA substrates were synthesized by IDT and purified as described above. Radiolabeled RNA substrates were prepared by 5'-end-labeling with T4 PNK (NEB) in the presence of gamma $^{32}$P-ATP. Radiolabeled DNA substrates were prepared by 5'-end-labeling with T4 PNK (NEB) in the presence of gamma $^{32}$P-ATP. For dsDNA substrates, non-target strand or target-strand was first 5'-end-labeled before annealing a 2-fold molar excess of the complementary strand at 95°C for 3 min in 1X hybridization buffer (20 mM Tris-Cl, pH 7.5, 150 mM KCl, 5 mM $MgCl_2$, 1 mM DTT) followed by slow-cooling to room temperature.

## Radiolabeled DNA cleavage assays

Cas12a-mediated DNA-cleavage assays were carried out in 1X cleavage buffer (20 mM Tris-Cl (pH 7.8 at 25°C), 150 mM KCl, 5 mM MgCl$_2$, 1% (v/v) glycerol) supplemented with 2 mM DTT or 1 mM TCEP. Radiolabeled DNA-cleavage assays to test mutated or truncated variants of AcrVA4 consisted of Cas12a, crRNA, $^{32}$P-labeled DNA substrate, and AcrVA4 at 30 nM, 36 nM, 1 nM, and 60 nM, respectively. Radiolabeled DNA-cleavage assays to test the effect of 5'-crRNA chemistry consisted of Cas12a, crRNA, $^{32}$P-labeled DNA substrate, and AcrVA4 at 30 nM, 36 nM, 0.5 nM, and 300 nM, respectively. In all cases, the RNP was formed at 37°C for 15 min before addition of AcrVA4 (unless otherwise indicated) and incubated at 37°C for 30 min. Reactions were initiated with the addition of target DNA at 37°C and samples quenched at 15 s, 30 s, 1 min, 10 min, and 30 min in a final concentration of 1.5X formamide loading buffer (2X concentration: 90% (v/v) formamide, 30 mM EDTA, 0.2% (w/v) SDS, 400 μg mL$^{-1}$ Heparin, and 0.5% (w/v) bromophenol blue) for 3 min at 95°C. Samples were resolved by 12% (v/v) urea-denaturing PAGE (0.5X TBE) and visualized by phosphoroimaging (Amersham Typhoon, GE Healthcare). The fraction of DNA cleavage was calculated as a ratio of the intensity of the product band relative to the total intensity of both the product and uncleaved DNA normalized to the background within each measured substrate in ImageQuant TL Software (GE Healthcare). Apparent rates were calculated by a fit to either a single or two phase exponential decay (Prism7, GraphPad). The rates with their associated standard deviations are included in the figure legends ($n = 3$ independent experiments).

## Phylogenetic analysis

A multiple sequence alignment of the full Cas12a and ancestral TnpB proteins (*Knott et al., 2019*; *Yan et al., 2019*) was generated using MAFFT v7.407 (`–localpair –maxiterate 1000`) (*Katoh and Standley, 2013*) (*Supplementary file 2*). A phylogenetic tree was constructed for the full proteins using IQTREE v1.6.6 (*Nguyen et al., 2015*) automatic model selection and 1000 bootstrap samplings. The tree was visualized using iTOL v.3 (*Letunic and Bork, 2016*). The RuvC domains for each protein in the alignment were inferred from the known Cas12a crystal structures (*Yamano et al., 2016*; *Yamano et al., 2017*; *Swarts and Jinek, 2019*) and a second alignment and phylogenetic tree were also produced from the RuvC domains for comparison.

## Size-exclusion chromatography and coupled di-angle light scattering

All experiments were run in 20 mM HEPES.K (pH 7.5), 200 mM KCl, 1 mM TECP, 1 mM MgCl$_2$) on a Superdex 10/300 Increase column (GE Healthcare) at 0.5 mL min$^{-1}$ using the Infinity 1260 Bio-SEC with light scattering module (Agilent). Light scattering was collected at 15° and 90° using a 658 nm laser. The system was calibrated using a 2 mg/mL BSA and dn/dc of 0.185. Calibration constants were determined as: 280 nm UV = 567.9, LS 90°=39111.5 and LS 15°=29921.9. LS 15° data were not used in our calculations. Cas12a and AcrVA4 concentrations were determined by nanodrop before combination with nucleic acid substrates and used as manual inputs for the mass calculation and a dn/dc of 0.185. All masses were determined using a first degree fit over the linear region of mass estimates for each peak using the Bio-SEC software V A.02.01 (Agilent).

## SEC-SAXS data collection, solution structure modeling, and analysis

AcrVA4 and AcrVA4 (Δ1–134) were prepared at ~5 mg/mL in 20 running buffer (20 mM HEPES-K pH 7.5, 200 mM KCl, 1 mM MgCl$_2$, 1 mM TCEP, 1% Glycerol). The LbCas12a-crRNA complex was prepared at 3.6 mg/mL by incubating LbCas12a with a 1.5 molar excess of crRNA at 37°C for 15 min before storage on ice and subsequent data collection. The LbCas12a-crRNA-AcrVA4 and LbCas12a-crRNA-AcrVA4 Δ1–134 complexes were prepared by first incubating LbCas12a with crRNA as described above. Then, a five-fold molar excess of AcrVA4 (or AcrVA4 Δ1–134) was added and incubated for a further 15 min at 37°C after which samples were stored on ice before data collection.

Small-angle x-ray scattering in-line with size-exclusion chromatography (SEC-SAXS) was collected at the SIBYLS beamline (bl12.3.1), at the Advanced Light Source at the Lawrence Berkeley National Laboratory, Berkeley, California (*Classen et al., 2013*). X-ray wavelength was set at λ = 1.127 Å and the sample-to-detector distance was 2,105 mm, resulting in scattering vector q, ranging from 0.01 Å$^{-1}$ to 0.4 Å$^{-1}$. The scattering vector is defined as q = 4πsinθ/λ, where 2θ is the scattering angle. Data were collected using a Dectris PILATUS3 × 2M detector at 20°C and processed as previously

described (*Dyer et al., 2014*; *Hura et al., 2009*). Briefly, a SEC-SAXS flow cell was directly coupled with an Agilent 1260 Infinity HPLC system using a Shodex KW-803 column. The column was equilibrated with running buffer with a flow rate of 0.45 mL/min. 3 s X-ray exposures were collected continuously over the 30 minute SEC elution. The SAXS frames recorded prior to the protein elution peak were used to subtract the signal for SAXS frames across the elution peak. The corrected frames were investigated by radius of gyration $R_g$ derived by the Guinier approximation $I(q)=I(0) \exp(-q^2R_g^2/3)$ with the limits $qR_g <1.3$. The elution peak was mapped by comparing the integral of ratios to background and $R_g$ relative to the recorded frame using the program SCÅTTER. The frames in the regions of least $R_g$ variation were averaged and merged in SCÅTTER to produce the highest signal-to-noise SAXS curves for the corresponding elution peak. These merged SAXS curves were used for further SAXS analysis including solution structure modeling. The Guinier plot indicated an aggregation-free state of the protein (*Figure 5—figure supplement 1C–D*). The P(r) function was used to determine the maximal dimension of the macromolecule ($D_{max}$) and estimate inter-domain distances (*Figure 5*) (*Putnam et al., 2007*). P(r) functions were normalized based on the molecular weight (MW) of the assemblies, as determined by calculated volume-of-correlation, $V_c$ (*Rambo and Tainer, 2013*). The SAXS frames across multiple peaks of LbCas12a-crRNA-AcrVA4 sample were deconvoluted into two components by RAW (*Hopkins et al., 2017*) to eliminate content of larger oligomeric states in the SAXS signal. The cryo-EM structure of LbCas12a-crRNA, LbCas12a-crRNA-AcrVA4 were used to build atomistic model by adding missing residues in protein and RNA moieties by MODELER (*Fiser et al., 2000*). These atomistic models were fit to the SAXS curves by FoXS (*Schneidman-Duhovny et al., 2013*; *Schneidman-Duhovny et al., 2016*). BILBOMD (*Pelikan et al., 2009*) approach was used to optimized conformation of AcrVA4 flexible tail regions.

## SEC-MALS data collection and analysis

Eluent was subsequently split (4 to 1) between the SAXS line and a multiple wavelength detector (UV-vis) at 280 nm, multi-angle light scattering (MALS), and refractometer. MALS experiments were performed using an 18-angle DAWN HELEOS II light scattering detector connected in tandem to an Optilab refractive index concentration detector (Wyatt Technology). System normalization and calibration was performed with bovine serum albumin using a 55 µL sample at 6.7 mg/mL in the same SEC running buffer and a dn/dc value of 0.185 mL/g. The MALS data were used to compliment the MWs calculated by the SAXS analysis and being furthest downstream, the MALS peaks were used to align the SAXS and UV-vis peaks along the x-axis (elution volume in mL/min) to compensate for fluctuations in timing and band broadening (*Figure 5—figure supplement 1A–B*). MALS, and differential refractive index data were analyzed using Wyatt Astra seven software to monitor the homogeneity of the sample molecular weights across the elution peak complementary to the above-mentioned SEC-SAXS signal validation (*Figure 5—figure supplement 1A–B*).

## Plasmid and *E. coli* strain construction for bacteriophage plaque assays

*E. coli* NEB 10-beta (New England Biolabs, #C3020K) was co-transformed with various combinations of two plasmids, the first encoding CRISPR-Cas machinery and the second encoding an anti-CRISPR protein. Each Cas12a ortholog was cloned with its minimal cognate CRISPR array (repeat-spacer-repeat) into a single expression vector (CamR, p15A origin). Cas12a was encoded under transcriptional control of the anhydrotetracycline (aTc)-inducible Tet promoter, while CRISPR arrays were transcribed from the strong constitutive, synthetic promoter known as proD. For each Cas12a variant, a spacer targeting the coding sequence of bacteriophage lambda Cro was cloned into the CRISPR array to facilitate phage interference, while a non-targeting, randomized spacer was cloned as a negative control. Anti-CRISPR proteins were cloned into a compatible vector (AmpR, SC101 origin) under transcriptional control of the arabinose-inducible pBAD promoter. AcrVA1 (cloned from Addgene, #115660) and AcrIIA4 (cloned from Addgene, #86836) were used as positive and negative anti-CRISPR controls, respectively.

## Bacteriophage production

Bacteriophage stocks for this study were generated from a lysogen of heat-inducible phage lambda (MG1655 λ+(cl857 *bor:kanR*)) through a modified liquid scaling method. Briefly, overnight cultures of lysogen were inoculated at an $OD_{600}$ of 0.02 into 25 mL of prewarmed 30°C LB Broth Lennox

media supplemented with 50 mg mL$^{-1}$ kanamycin sulfate and cultured at 30°C, 180 RPM to an OD$_{600}$ of 0.2. The culture was then transferred to a 37°C water bath and shaken for 2 hr at 180 RPM to facilitate phage lysis. Phage stock solution was collected by 15 min exposure to chloroform, followed by centrifugation (8000 RPM, 10 min), and supernatant passed through a 0.22 μm sterile filter (SCGP00525). Lysates were stored at 4°C until use.

## Bacteriophage plaque assays

All liquid, solid, and soft media were prepared with LB Broth Lennox (1% (w/v) tryptone, 0.5% (w/v) yeast extract, 0.5% (w/v) NaCl) and supplemented with 34 mg mL$^{-1}$ chloramphenicol and 100 mg mL$^{-1}$ carbenicillin to select for CRISPR and anti-CRISPR plasmids, respectively. Bottom and top agar were prepared with 1.5% (w/v) and 0.7% (w/v) agar, respectively. All agar media used for plaque assays were supplemented with inducer for full anti-CRISPR induction (0.1% (w/v) L-arabinose). Agar for all CRISPR effector constructs was supplemented with sufficient inducer to observe full CRISPR-Cas effector activity (2 nM for wild-type AsCas12a and LbCas12a, 14 nM for As*Cas12a, and 4 nM for Lb#Cas12a). *E. coli* NEB 10-beta strains harboring both a CRISPR plasmid and anti-CRISPR plasmid were streaked onto LB agar plates without inducers and grown overnight at 37°C. All strains were inoculated from LB agar plates into LB Broth Lennox containing appropriate antibiotics and shaken at 37°C overnight before performing plaque assays. 100 μL of overnight culture was mixed with 5 mL top agar and overlaid onto 5 mL bottom agar and allowed to completely solidify. Ten-fold serial dilutions of heat-inducible phage lambda (λ cl857 *bor:kanR*) were prepared in SM buffer (Teknova, #S2210). 2 μL of each serial dilution was spotted onto the top agar, and allowed to dry before incubating at 37°C overnight. All experiments were performed on three different days with three independent bacterial cultures.

## Mammalian cell culture

All mammalian cell cultures were maintained in a 37°C incubator at 5% CO$_2$. HEK293T (293FT; Thermo Fisher Scientific, #R70007) human embryonic kidney cells and derivatives thereof were grown in Dulbecco's Modified Eagle Medium (DMEM; Corning Cellgro, #10–013-CV) supplemented with 10% fetal bovine serum (FBS; Seradigm, #1500–500), and 100 Units/ml penicillin and 100 mg/ml streptomycin (100-Pen-Strep; GIBCO #15140–122). HEK293T cells were tested for absence of mycoplasma contamination (UC Berkeley Cell Culture facility) by fluorescence microscopy of methanol fixed and Hoechst 33258 (Polysciences #09460) stained samples.

## Lentiviral transduction

Lentiviral particles were produced in HEK293T cells using polyethylenimine (PEI; Polysciences, #23966) based transfection of plasmids. HEK293T cells were split to reach a confluency of 70–90% at time of transfection. Lentiviral vectors were co-transfected with the lentiviral packaging plasmid psPAX2 (Addgene, #12260) and the VSV-G envelope plasmid pMD2.G (Addgene, #12259). Transfection reactions were assembled in reduced serum media (Opti-MEM; GIBCO, #31985–070). For lentiviral particle production in 6-well plates, 1 μg transfer vector, 0.5 μg psPAX2 and 0.25 μg pMD2.G were mixed in 400 μl Opti-MEM, followed by addition of 5.25 μg PEI. After 20–30 min incubation at room temperature, the transfection reactions were dispersed over the HEK293T cells. Media were changed 12–18 hr post-transfection, and virus harvested at 42–48 hr post-transfection. Viral supernatants were filtered using 0.45 μm polyethersulfone (PES) membrane filters, diluted in cell culture media if appropriate, and added to target cells. Polybrene (5 μg/ml; Sigma-Aldrich, #H9268) was supplemented to enhance transduction efficiency, when appropriate. Transduced target cell populations (HEK293T) were usually selected 24–48 hr post-transduction using puromycin (1.0 μg/ml; InvivoGen, #ant-pr-1) or hygromycin B (400 μg/ml; Thermo Fisher Scientific, #10687010).

## Lentiviral vectors

All-in-one doxycycline-inducible Tet-On lentiviral vectors expressing various anti-CRISPRs (Acrs) were cloned based on LT3GEPIR (*Fellmann et al., 2013*). In brief, LT3GEPIR was digested with BamHI-HF/EcoRI-HF to replace the intervening GFP-miR-E cassette with GFP alone or GFP-P2A-Acr polycistronic constructs using Gibson assembly. This yielded pCF570 (GFP), pCF571 (GFP-P2A-AcrVA1),

pCF572 (GFP-P2A-AcrVA4), pCF573 (GFP-P2A-AcrVA5), pCF574 (GFP-P2A-AcrIIA4), and pCF575 (GFP-P2A-AcrVA4-Delta-1–134).

## RNP assembly for mammalian genome editing assays

AsCas12a (purified here; IDT), AsCas12a-chimera (purified here) and SpCas9 (IDT) RNPs were prepared in low-binding PCR tubes. For Cas9, crRNA (IDT) and tracrRNA (IDT) were pre-annealed (1:1 mix), followed by dilution in reaction buffer (IDT), according to manufacturer procedures. For AsCas12a, crRNAs (IDT) were diluted in reaction buffer (IDT). RNP complexes were then assembled by mixing corresponding guide RNAs with Cas proteins (1.1:1 mix) to obtain 10 µM RNP solutions.

## Inducible anti-CRISPR (Acr) genome editing reporter assay

To establish a rapid assay for reliable quantification of genome editing efficiency of diverse Cas enzymes in the presence of select anti-CRISPRs (Acrs), we built a fluorescence-based reporter assay with doxycycline-inducible Acr expression. Specifically, we generated a monoclonal HEK293T-based reporter cell line stably expressing a polycistronic Hygro-P2A-mCherry construct by transducing HEK293T cells with the lentiviral vector pCF525-EF1a-Hygro-P2A-mCherry (Addgene, #115796) (*Watters et al., 2018*), followed by hygromycin selection and isolation of monoclonal cell lines. Twelve mCherry-positive clones were identified by fluorescence imaging and further assessed for homogeneous morphology as well as percentage mCherry+ and median fluorescence intensity (MFI) by flow cytometry (*Figure 6—figure supplement 4A*). HEK-RC1 cells (HEK293T-pCF525 reporter mCherry clone 1) are derived from the clone that performed best in these tests.

For doxycycline-inducible Acr expression, HEK-RC1 were stably transduced at single-copy (<5% initial transduction efficiency) with the lentiviral constructs pCF570 (GFP), pCF571 (GFP-P2A-AcrVA1), pCF572 (GFP-P2A-AcrVA4), pCF573 (GFP-P2A-AcrVA5), pCF574 (GFP-P2A-AcrIIA4) and pCF575 (GFP-P2A-AcrVA4-Delta-1–134), followed by selection on puromycin. All cell lines were then tested for doxycycline (1 µg/ml; Sigma-Aldrich) inducible expression of GFP or the GFP-P2A-Acr constructs.

To identify suitable guide RNAs for AsCas12a and SpCas9 RNP-based editing of the Hygro-P2A-mCherry polycistronic construct expressed in HEK-RC1 reporter cell lines, we designed and synthesized four AsCas12a crRNAs (21-mers, IDT) and four SpCas9 sgRNAs (20-mers, dual-guide RNA system, IDT). The target sequences for these guide RNAs are: cr-mCherry-1 (#058; TTCTGCATTACGGGGCCGTCG), cr-Hygro-1 (#059; tgtacgcccgacagtcccggc), cr-Hygro-2 (#060; cactatcggcgagtacttcta), cr-Hygro-3 (#061; gatgatgcagcttgggcgcag), sg-mCherry-1 (#062; GTGATGAACTTCGAGGACGG), sg-mCherry-2 (#063; CAAGTAGTCGGGGATGTCGG), sg-Hygro-1 (#064; GGGGCGTCGGTTTCCACTAT), and sg-Hygro-2 (#065; AGATGTTGGCGACCTCGTAT). RNPs were assembled and transfected into HEK-RC1 reporter cells using Lipofectamine 2000 (Thermo Fisher Scientific) according to manufacturer procedures, with 100 pmol RNP per sample in 24-well plates with 1 ml cell culture medium. In brief, per sample, 100 pmol RNP at 10 µM was diluted in 25 µl Opti-MEM, and 1.6 µl Lipofectamine 2000 was diluted in 25 µl Opti-MEM in a separate tube. Diluted RNP was then added to diluted transfection reagent, incubated for 15 min, and added to cells. Flow cytometry-based quantification of mCherry fluorescence in HEK-RC1 edited with various AsCas12a crRNA and SpCas9 sgRNA RNPs revealed strong editing differences among AsCas12a RNPs, while all tested SpCas9 RNPs worked well (*Figure 6—figure supplement 4B*). We chose AsCas12a cr-Hygro-3 (#061) and SpCas9 sg-Hygro-1 (#064) for all further assays.

## Quantification of inhibition using the acr genome editing reporter assay

HEK-RC1 Acr and control cell lines (HEK-RC1-pCF570/571/572/573/574/575) were seeded 36 hr prior to transfection and treated with doxycycline (1 µg/ml; Sigma-Aldrich) to induce expression of the Acr and control constructs (Tet-On system). Cells were then transfected at approximately 30% confluency with 100 pmol RNP per sample, in triplicates, using Lipofectamine 2000 (as described above). At 24 hr post-transfection, cells were split for further growth and doxycycline (1 µg/ml) renewed. At day four post-transfection, non-transfected and RNP-transfected cells were analyzed by flow cytometry to quantify loss of mCherry fluorescence (editing) in the GFP-positive populations (expressing the Acr and control constructs). Per sample, 10,000–30,000 events were acquired (Attune NxT, Thermo Fisher Scientific). Propagation of uncertainty was taken into consideration

when reporting data and their uncertainty (standard deviation) as functions of measurement variables. Unless otherwise noted, error bars indicate the standard deviation of triplicates, and significance was assessed by comparing samples to their respective controls using unpaired, two-tailed t tests (alpha = 0.05).

## Acknowledgements

The authors acknowledge financial support from the Defense Advanced Research Projects Agency (DARPA) (award HR0011-17-2-0043 to JAD), the Paul G Allen Frontiers Group, and the National Science Foundation (MCB-1244557 to JAD). JAD is an investigator of the Howard Hughes Medical Institute (HHMI), and this study was supported in part by HHMI; JAD is also a Paul Allen Distinguished Investigator. GJK was funded in part by the William M Keck foundation. J-JL was supported by the National Institutes of Health under award number P50GM082250 (HARC Center) and is a Pfizer Fellow of the Life Sciences Research Foundation. BAS was supported by the National Science Foundation Graduate Research Fellowship (No. DGE 1752814). MJL was supported in part by the Centers for Excellence in Genomic Science of the National Institutes of Health under award number RM1HG009490. CF was supported by a US NIH Pathway to Independence Award (R00GM118909) from the National Institute of General Medical Sciences (NIGMS). Small-angle X-ray scattering experiments were conducted at the Advanced Light Source (ALS) beamline 12.3.1. SAXS data collection at SIBYLS is funded through DOE BER Integrated Diffraction Analysis Technologies (IDAT) program and NIGMS grant P30 GM124169-01, ALS-ENABLE. This work used NE-CAT beamlines (GM124165), a Pilatus detector (RR029205), and an Eiger detector (OD021527) at the APS (DE-AC02-06CH11357). We thank Drew Endy (Stanford University, CA) for the generous gift of λ cI857 *bor::kanR*. Schematics for a subset of images created with Biorender.com.

## Additional information

### Competing interests

Gavin J Knott and Basem Al-Shayeb, Jun-Jie Liu, Brittney W Thornton: The Regents of the University of California have patents pending for CRISPR technologies on which the authors are inventors. Christof Fellmann: The Regents of the University of California have patents pending for CRISPR technologies on which the authors are inventors. CF is a co-founder of Mirimus, Inc. Jennifer A Doudna: The Regents of the University of California have patents pending for CRISPR technologies on which the authors are inventors. JAD is a co-founder of Caribou Biosciences, Editas Medicine, Intellia Therapeutics, Scribe Therapeutics, and Mammoth Biosciences. JAD is a scientific advisory board member of Caribou Biosciences, Intellia Therapeutics, eFFECTOR Therapeutics, Scribe Therapeutics, Synthego, Metagenomi, Mammoth Biosciences, and Inari. JAD is a Director at Johnson & Johnson and has sponsored research projects by Pfizer, Roche Biopharma, and Biogen. The other authors declare that no competing interests exist.

### Funding

| Funder | Grant reference number | Author |
| --- | --- | --- |
| Defense Advanced Research Projects Agency | HR0011-17-2-0043 | Jennifer A Doudna |
| Paul G. Allen Frontiers Group | | Jennifer A Doudna |
| National Science Foundation | MCB-1244557 | Jennifer A Doudna |
| Howard Hughes Medical Institute | | Jennifer A Doudna |
| National Institutes of Health | P50GM082250 | Jun-Jie Liu |
| National Science Foundation | DGE 1752814 | Basem Al-Shayeb |
| National Institutes of Health | RM1HG009490 | Marco J Lobba |
| National Institute of General Medical Sciences | R00GM118909 | Christof Fellmann |

| W. M. Keck Foundation | Gavin J Knott |
| --- | --- |

The funders had no role in study design, data collection and interpretation, or the decision to submit the work for publication.

## Author contributions
Gavin J Knott, Conceptualization, Data curation, Formal analysis, Supervision, Validation, Investigation, Visualization, Methodology, Writing—original draft, Project administration, Writing—review and editing; Brady F Cress, Formal analysis, Validation, Investigation, Visualization, Methodology, Writing—review and editing; Jun-Jie Liu, Basem Al-Shayeb, Resources, Formal analysis, Investigation, Visualization, Methodology, Writing—review and editing; Brittney W Thornton, Resources, Formal analysis, Validation, Investigation, Visualization, Writing—review and editing; Rachel J Lew, Resources, Validation, Investigation; Daniel J Rosenberg, Michal Hammel, Resources, Data curation, Software, Formal analysis, Investigation, Visualization, Methodology, Writing—review and editing; Benjamin A Adler, Validation, Investigation, Methodology, Writing—review and editing; Marco J Lobba, Resources, Software, Formal analysis, Investigation, Methodology; Michael Xu, Resources, Investigation; Adam P Arkin, Resources, Supervision, Funding acquisition; Christof Fellmann, Resources, Formal analysis, Supervision, Investigation, Visualization, Methodology, Writing—review and editing; Jennifer A Doudna, Resources, Supervision, Funding acquisition, Project administration, Writing—review and editing

## Author ORCIDs
Gavin J Knott  https://orcid.org/0000-0002-9007-6273
Daniel J Rosenberg  http://orcid.org/0000-0001-8017-8156
Adam P Arkin  http://orcid.org/0000-0002-4999-2931
Christof Fellmann  https://orcid.org/0000-0002-9545-5723
Jennifer A Doudna  https://orcid.org/0000-0001-9161-999X

## Decision letter and Author response
Decision letter https://doi.org/10.7554/eLife.49110.036
Author response https://doi.org/10.7554/eLife.49110.037

# Additional files

## Supplementary files
• Supplementary file 1. SDS-PAGE of purified proteins used in this study. Purified proteins used in this study.
DOI: https://doi.org/10.7554/eLife.49110.028
• Supplementary file 2. Cas12a multiple sequence alignment. Multiple sequence alignment of Cas12a proteins as described in Materials and methods.
DOI: https://doi.org/10.7554/eLife.49110.029
• Transparent reporting form
DOI: https://doi.org/10.7554/eLife.49110.030

## Data availability
The cryo-EM map of the LbCas12a-crRNA-AcrVA4 complex at 2.99 Å resolution (State I) and the refined coordinates model have been deposited to the EMDB and PDB with accession codes EMD-20266 and PDB-6P7M, respectively. The cryo-EM map of the 2[LbCas12a-crRNA-AcrVA4] complex at 4.91 Å resolution (State II) and the refined coordinates have been deposited to the EMDB and PDB with accession codes EMDB-20267 and PDB-6P7N, respectively. All other data generated or analyzed during this study are included in the manuscript or supporting files.

The following datasets were generated:

| Author(s) | Year | Dataset title | Dataset URL | Database and Identifier |
|---|---|---|---|---|
| Knott, GJ, Liu, JJ, Doudna, JA | 2019 | LbCas12a-crRNA-AcrVA4 State I | https://www.rcsb.org/structure/6P7M | RCSB Protein Data Bank, 6P7M |
| Knott, GJ, Liu, JJ, Doudna, JA | 2019 | LbCas12a-crRNA-AcrVA4 State II | https://www.rcsb.org/structure/6P7N | RCSB Protein Data Bank, 6P7N |

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
