## [Decision Letter]

Thank you for submitting your article "Structural switching of CRISPR-Cas12a inhibition specificity" for consideration by *eLife*. Your article has been reviewed by Cynthia Wolberger as the Senior Editor, a Reviewing Editor, and two reviewers. The following individuals involved in review of your submission have agreed to reveal their identity: Hong-Wei Wang (Reviewer #3).

The reviewers have discussed the reviews with one another and the Reviewing Editor has drafted this decision to help you prepare a revised submission.

Summary:

Knott and colleagues characterize the interactions of the Cas12a RNP with its inhibitor AcrVA4, one of an emerging class of molecules that can suppress CRISPR mediated bacterial defense systems. AcrVA4 is interesting in being able to inactivate Cas12a from some species, such as *L. bacterium*, but not others. The authors present two cryo-EM structures of the LbCas12a/crRNA/AcrVA4 complex, termed State I and State II. State 1 includes density for one copy of Cas12a and one well resolved copy of AcrVA4, while State II represents the heterotetrameric complex. Interestingly while AcrVA4 is known to homodimerize, the authors show that dimerization of AcrVA4 is not required for Cas12a inhibition in vitro or in vivo. Within the complex, AcrVA4 makes specific interactions with the 5' end of the mature Crispr RNA and with the WED, Rec and RuvC domains of Cas12a. The structure points to an allosteric interaction of AcrVA4 blocking the conformational change needed for R-loop formation as the mechanism of Cas12a inhibition. The authors also examine the AsCas12a protein that is not inhibited by AcrVA4, and show that this is due to an extra two helix bundle in the AsCas12a preventing AcrVA4 interaction with the AsCas12a WED domain. All of the above interactions are tested with a variety of point mutations, single domain proteins, and chimeric constructs.

The two reviewers both found this manuscript to be well written and the study well supported through multiple experimental approaches. The work provides important new insights into the biology of CRISPR defense systems and has technological implications for the design of new genome editing tools. With some revision the paper is an appropriate and interesting publication for *eLife*.

Essential revisions:

1) In Figure 2 the authors nicely show that the 5' end of the crRNA is critical for AcrVA4 inhibition. Does the 5' end of the crRNA influence AcrVA4 recruitment to LbCas12a? Is the loss in inhibition due to a binding defect?

2) There is no quality control shown for any of the AcrVA4 mutants. The authors should include some type of quality control (SEC curves, SEC-MALS, etc.) to ensure that the AcrVA4 mutants are stable, properly folded, and have the appropriate oligomeric state.

3) It is not always clear whether the effects described were the only ones occurring/tested or the only ones being presented. For example, the logic is not clear as to why the authors chose W178 of AcrVA4 to test effects on LbCas12a dsDNA cis-cleavage. As seen in Figure 4, there seem to be other residues involved in the interaction between AcrVA4 and the BH. Were other amino acids in AcrVA4 also tested? Similarly, in addition to the conformational changes of BH, do any other domains exhibit conformational rearrangements when the AcrVA4 binds to the LbCas12a-crRNA?

4) Did the authors try to apply a C2 symmetry to the dimeric State II structure during the refinement to improve the resolution or map quality?

5) There are lots of acronyms in this paper that are hard to keep track of. It would be useful if the authors could define the acronyms in the figure legends and not just the main text.

6) In Figure 4—figure supplement 4 it is difficult to distinguish between the alternative loop conformations.

7). In Figure 5F it looks as if there is a small difference in phage survival with the delta1-134 mutant, suggesting a minor role for dimerization in vivo. The authors should comment on this.

8) Figure 6A is difficult to see and should be enlarged and AcrVA4 should be shown as a transparent surface.

9) EM reconstructions cannot be described as "electron density". Electron density is determined by X-ray crystallography. The authors should replace "electron density" with "EM density" throughout the manuscript.

---

## [Author Response]

Summary:Knott and colleagues characterize the interactions of the Cas12a RNP with its inhibitor AcrVA4, one of an emerging class of molecules that can suppress CRISPR mediated bacterial defense systems. […] The two reviewers both found this manuscript to be well written and the study well supported through multiple experimental approaches. The work provides important new insights into the biology of CRISPR defense systems and has technological implications for the design of new genome editing tools. With some revision the paper is an appropriate and interesting publication for eLife.

The authors thank the reviewers for their evaluation of this work. We have made a number of minor changes to the text and figures to improve the clarity of the narrative, included additional experiments that further support our conclusions, and directly addressed the reviewer comments.

Essential revisions:1) In Figure 2 the authors nicely show that the 5' end of the crRNA is critical for AcrVA4 inhibition. Does the 5' end of the crRNA influence AcrVA4 recruitment to LbCas12a? Is the loss in inhibition due to a binding defect?

Given the extent of the interface between AcrVA4 and the WED domain of LbCas12a, we hypothesize that the presence of any steric bulk in place of the mature crRNA 5’-OH (such as the tested 5’ PO_4_ or an uncleavable pre-crRNA) would result in a loss of binding. To directly address this question, we carried out size exclusion chromatography to assay binding of AcrVA4 to an LbCas12a RNP in the presence of varied crRNA 5’ chemistry. Consistent with our hypothesis, the presence of a 5’PO_4_ or an uncleavable pre-crRNA reduced formation of the LbCas12a-crRNA-AcrVA4 complex. These data are now described within the Results section as related to Figure 2—figure supplement 2B.

2) There is no quality control shown for any of the AcrVA4 mutants. The authors should include some type of quality control (SEC curves, SEC-MALS, etc.) to ensure that the AcrVA4 mutants are stable, properly folded, and have the appropriate oligomeric state.

We thank the reviewer for this comment and agree it is pertinent to include data describing the solution state of the AcrVA4 mutants. All of the AcrVA4 mutants described in this study eluted from the SEC at volumes comparable to wild-type AcrVA4, consistent with appropriate folding and oligomeric state. We now include the SEC traces for each AcrVA4 point mutant as Figure 2—figure supplement 2A. We also include an SDS-PAGE gel for each purified protein described in this study (Supplementary File 1).

3) It is not always clear whether the effects described were the only ones occurring/tested or the only ones being presented. For example, the logic is not clear as to why the authors chose W178 of AcrVA4 to test effects on LbCas12a dsDNA cis-cleavage. As seen in Figure 4, there seem to be other residues involved in the interaction between AcrVA4 and the BH. Were other amino acids in AcrVA4 also tested? Similarly, in addition to the conformational changes of BH, do any other domains exhibit conformational rearrangements when the AcrVA4 binds to the LbCas12a-crRNA?

For AcrVA4 point mutants presented in Figure 2, these were selected by examining the interface for residues contacting LbCas12a or the bound crRNA (~ 3.0-Å cut-off) in addition to those buried within the interface. Additionally, a number of the contacts between AcrVA4 and LbCas12a are formed by main chain interactions. We now describe this clearly within the text. With respect to W178, this residue was selected because it dominates the interface between AcrVA4 and the Cas12a bridge-helix (BH). We have updated the Results section to better explain our logic for mutagenesis.

4) Did the authors try to apply a C2 symmetry to the dimeric State II structure during the refinement to improve the resolution or map quality?

We did attempt refinement with C2 symmetry applied, but this did not improve the resolution or quality of the cryo-EM map, likely due to the particles not having perfect C2 symmetry.

5) There are lots of acronyms in this paper that are hard to keep track of. It would be useful if the authors could define the acronyms in the figure legends and not just the main text.

We agree with the reviewer and have updated the main text to limit acronyms where possible in addition to defining them within figure legends.

6) In Figure 4—figure supplement 4 it is difficult to distinguish between the alternative loop conformations.

This figure panel has been merged with Figure 4—figure supplement 1 and the image updated to better distinguish the two conformations.

7). In Figure 5F it looks as if there is a small difference in phage survival with the delta1-134 mutant, suggesting a minor role for dimerization in vivo. The authors should comment on this.

We have included a sentence in the Results section highlighting the small phenotypic difference in plaques for the truncation relative to full-length AcrVA4. Additionally, we updated the discussion to comment on stability and expression level of the wild-type versus truncated AcrVA4 as a potential source for this difference.

8) Figure 6A is difficult to see and should be enlarged and AcrVA4 should be shown as a transparent surface.This has been updated.9) EM reconstructions cannot be described as "electron density". Electron density is determined by X-ray crystallography. The authors should replace "electron density" with "EM density" throughout the manuscript.

This has been corrected throughout the text.